# FROM STABILITY TO CHAOS: ANALYZING GRADIENT DESCENT DYNAMICS IN QUADRATIC REGRESSION

## ABSTRACT

We conduct a comprehensive investigation into the dynamics of gradient descent using large-order constant step-sizes in the context of quadratic regression models. Within this framework, we reveal that the dynamics can be encapsulated by a specific cubic map, naturally parameterized by the step-size. Through a fine-grained bifurcation analysis concerning the step-size parameter, we delineate five distinct training phases: (1) monotonic, (2) catapult, (3) periodic, (4) chaotic, and (5) divergent, precisely demarcating the boundaries of each phase. As illustrations, we provide examples involving phase retrieval and two-layer neural networks employing quadratic activation functions and constant outer-layers, utilizing orthogonal training data. Our simulations indicate that these five phases also manifest with generic non-orthogonal data. We also empirically investigate the generalization performance when training in the various non-monotonic (and non-divergent) phases. In particular, we observe that performing an ergodic trajectory averaging stabilizes the test error in non-monotonic (and non-divergent) phases.

## 1 INTRODUCTION

Iterative algorithms like the gradient descent and its stochastic variants are widely used to train deep neural networks. For a given step-size parameter $\eta > 0$, the gradient descent algorithm is of the form $w^{(t+1)} = w^{(t)} - \eta \nabla \ell(w^{(t)})$ where $\ell$ is the training objective function being minimized, which depends on the loss function and the neural network architecture and the dataset. Classical optimization theory operates under small-order step-sizes. In this regime, one can think of the gradient descent algorithm as a discretization of so-called gradient flow equation given by $\dot{w}^{(t)} = -\nabla \ell(w^{(t)})$, which could be obtained from the gradient descent algorithm by letting $\eta \to 0$. Additionally, assuming that the objective function $\ell$ has gradients that are $L$-Lipschitz, selecting a step-size $\eta < 1/L$ guarantees convergence to stationarity.

In stark contrast to traditional optimization, recent empirical studies in deep learning have revealed that training deep neural networks with large-order step-sizes yields superior generalization performance. Unlike the scenario with small step-sizes, where gradient descent dynamics follow a monotonic pattern, larger step-sizes introduce a more intricate behavior. Various patterns like catapult (also related to *edge of stability*), periodicity and chaotic dynamics in neural network training with large step-sizes have been observed empirically, for example, by Lewkowycz et al. (2020), Jastrzebski et al. (2020), Cohen et al. (2021), Lobacheva et al. (2021), Gilmer et al. (2022), Zhang et al. (2022), Kodryan et al. (2022), Herrmann et al. (2022). In fact, the necessity for larger step-sizes to expedite convergence and the ensuing chaotic behavior has also been observed empirically outside the deep learning community by Van Den Doel and Ascher (2012), much earlier.

Faster convergence of gradient descent with iteration-dependent step-size schedules that have specific patterns (including cyclic and fractal patterns) has been examined empirically by Lebedev and Finogenov (1971); Smith (2017); Oymak (2021); Agarwal et al. (2021); Goujaud et al. (2022); Grimmer (2023), with Altschuler and Parrilo (2023) and Grimmer et al. (2023) proving the state-of-the art remarkable results; see also Altschuler and Parrilo (2023, Section 1.2) for a historical overview. Notable, the stated faster convergence behavior of gradient descent requires large order step-sizes, very much violating the classical case. More importantly, the corresponding optimization trajectory, while being non-monotonic, exhibits intriguing patterns (Van Den Doel and Ascher, 2012).

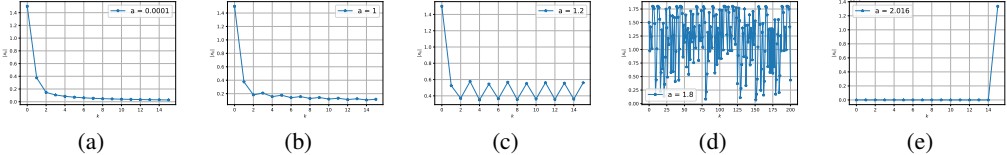

(a)         (b)         (c)         (d)         (e)

Figure 1: Phases of cubic-map based dynamical system in (2.1) parameterized by $a$. Sub-figure 1(a) corresponds to the monotonic phases, where the dynamics monotonically decays to zero. Sub-figure 1(b) corresponds to the catapult phase where the dynamics decays to zero but is non-monotonic. Sub-figure 1(c) corresponds to the periodic phase, where the dynamics decays and settles in a period-2 orbit (i.e., shuttles between two points) but never decays to zero. Sub-figures 1(d) and 1(e) correspond to the chaotic phase (see Definition 1) and divergent phases, respectively. Note that the order of $x$-axis and $y$-axis in Sub-figures 1(d) and 1(e) are different from the rest.

Considering the aforementioned factors, gaining insight into the dynamics of gradient descent with large-order step-sizes emerges as a pivotal endeavor. A precise theoretical characterizing of the gradient descent dynamics in the large step-size regime for deep neural network, and other such non-convex models, is a formidably challenging problem. Existing findings (as detailed in Section 1.1) often rely on strong assumptions, even when attempting to delineate a subset of the aforementioned patterns, and do not provide a comprehensive account of the entire narrative underlying the training dynamics. Recent research, such as Agarwala et al. (2023), Zhu et al. (2022), and Zhu et al. (2023b), has pivoted towards comprehending the dynamics of quadratic regression models based on a *local* analysis. These models offer a valuable testing ground due to their ability to provide tractable approximations for various machine learning models, including phase retrieval, matrix factorization, and two-layer neural networks, all of which exhibit unstable training dynamics. Despite their seeming simplicity, a fine-grained understanding of their training dynamics is far from trivial. Building in this direction, the primary aim of our work is to attain a precise characterization of the training dynamics of gradient descent in quadratic models, thereby fostering a deeper comprehension of the diverse phases involved in the training process.

> **Contribution 1.** We perform a *fine-grained, global theoretical analysis* of a cubic-map-based dynamical system (see Equation 2.1), and identify the precise boundaries of the following five phases: (i) monotonic, (ii) catapult, (iii) periodic, (iv) Li-Yorke chaotic, and (v) divergent. See Figure 1 for an illustration, and Definition 2 and Theorem 2.1 for formal results. We show in Theorem 3.2 and 3.3, that the dynamics of gradient descent for two non-convex statistical problems, namely phase retrieval and two-layer neural networks with constant outer layers and quadratic activation functions, with orthogonal training data is captured by the cubic-map-based dynamical system. We provide empirical evidence of the presence of similar phases in training with non-orthogonal data.

We also empirically examine the effect of training models in the above-mentioned phases, in particular the non-monotonic ones, on the generalization error. Indeed, provable model-specific statistical benefits for training in catapult phase are studied in Lyu et al. (2022); Ahn et al. (2022b). Lim et al. (2022) proposed to induce controlled chaos in the training trajectory to obtain better generalization. Approaches to explain generalization with chaotic behavior are examined in Chandramoorthy et al. (2022) based on a relaxed notion of statistical algorithmic stability. Although our focus is on gradient descent, related notions of generalization of stochastic gradient algorithms, based on characterizing the fractal-like properties of the invariant measure they converge to (with larger-order constant step-size choices) have been explored, for example, in Birdal et al. (2021); Camuto et al. (2021); Dupuis et al. (2023); Hodgkinson et al. (2022). Hence, we also conduct empirical investigations into the performance of generalization when training within the different non-monotonic (and non-divergent) phases and make the following contribution.

> **Contribution 2**. We propose a natural ergodic trajectory averaging based prediction mechanism (see Section 4.2) to stabilize the predictions when operating in any non-monotonic (and non-divergent) phase.

## 1.1 RELATED WORKS

**Specific Models.** Zhu et al. (2023b) and Chen and Bruna (2023) studied gradient descent dynamics for minimizing the functions $\ell(u, v) = (u^2v^2 - 1)^2$ and $\ell = (u^2 - 1)^2$, respectively. Both works primarily focused on characterizing period-2 orbits and hint at the possibility of chaos without rigorous theoretical justifications. Furthermore, their proofs are relatively tedious and very different from ours. Song and Yun (2023) provided empirical evidence of chaos for minimizing $\ell(u, v) = $ using gradient descent. However, their results are not applicable to quadratic regression models. Ahn et al. (2022a) examined the Edge of Stability (EoS) between the monotonic and catapult phase for minimizing $\ell(u, v) = l(uv)$, where $l$ is convex, even, and Lipschitz. Their analysis is not directly extendable to the quadratic regression models we consider in this work. See also the discussion below Theorem 2.1 for important technical comparisons. Wang et al. (2022) analyzed additional benefits (e.g., taming homogeneity) of gradient descent with large step-sizes for matrix factorization.

Agarwala et al. (2023) explored gradient descent dynamics for a class of quadratic regression models and identified the EoS. Zhu et al. (2023a;b) also studied the catapult phase and EoS for a class of quadratic regression models. Agarwala and Dauphin (2023) examined the EoS in the context of Sharpness Aware Minimization for quadratic regression models. The above works are related to our work in terms of the model that they study. However, none of the above works characterize the five distinct phases (with precise boundaries) like we do, along with precise boundaries. Furthermore, our analysis is distinct (and is also global[1]) from the above works and is firmly grounded in the rich literature on dynamical systems.

**General results.** Lewkowycz et al. (2020) empirically examine the catapult phase, particularly in neural networks with one hidden layer and linear activations. Cohen et al. (2021) and Ahn et al. (2022b) provide insights into the EoS. Damian et al. (2023) propose self-stabilization as a phenomenological reason for the occurrence of catapults and EoS in gradient descent dynamics. Kreisler et al. (2023) investigate how gradient descent monotonically decreases the sharpness of Gradient Flow solutions, specifically in one-dimensional deep neural networks. Although they do not formally prove the existence of chaos in the dynamics, they conjecture its possibility. Arora et al. (2022) and Lyu et al. (2022) explore sharpness reduction flows, related to the above findings. Andriushchenko et al. (2023) prove that large step-sizes in gradient descent can lead to the learning of sparse features. Wu et al. (2023) investigate the EoS phenomenon for logistic regression. Kong and Tao (2020) theoretically explore the chaotic dynamics (and related stochasticity) in gradient descent for minimizing multi-scale functions under additional assumptions. While being extremely insightful, their results are fairly qualitative and are not directly applicable to the cubic maps analyzed in our work. As we focus on specific models, our results are more precise and quantitative.

**Dynamical systems.** Our results draw upon the rich literature available in the field of dynamical systems. We refer the interested reader to Alligood et al. (1997), Lasota and Mackey (1998), Devaney (1989), Ott (2002), De Melo and Van Strien (2012) for a book-level introduction. Birfurcation analysis of some classes of cubic maps has been studied, for example, by Skjolding et al. (1983), Rogers and Whitley (1983), Branner and Hubbard (1988) and Milnor (1992). Some of the above works are rather empirical, and the exact maps considered in the above works differ significantly from our case.

## 2 ANALYZING A DISCRETE DYNAMICAL SYSTEM WITH CUBIC MAP

**Notations and definitions.** We say a sequence $\{x_k\}_{k=0}^{\infty}$ is increasing (decreasing), if $x_{t+1} \geq x_t$ ($x_{t+1} \leq x_t$) for any $t$. Moreover, it is strictly increasing (decreasing) if the equalities never hold. For a real-valued function $f$ and a set $S$, define $f(S) = \{f(x) : x \in S\}$, and $f^{(k)}(x) := f(f^{(k-1)}(x))$ for any $k \in \mathbb{N}_+$ with $f^{(0)}(x) = x$. The preimage of $x$ under $f$ on $S$ is the set $f^{-1}(x) := \{y \in S : f(y) = x\}$. We say a property $P$ holds for almost every $x \in S$ or almost surely in $S$, if the subset $\{x \in S : \text{property } P \text{ does not hold for } x\}$ is Lebesgue measure zero. A critical point of $f$ is a point $x$ satisfying $f'(x) = 0$. We call $x_0$ a period-$k$ point of $f$, when $f^{(k)}(x_0) = x_0$ and $f^{(i)}(x_0) \neq x_0$ for any $0 \leq i \leq k - 1$. The orbit of a point $x_0$ denotes the sequence $\{f^{(t)}(x_0)\}_{t=0}^{\infty}$. A point $x_0$ is called asymptotically periodic if there exists a periodic point $y_0$ such that $\lim_{t \to \infty} |f^{(t)}(x_0) - f^{(t)}(y_0)| = 0$. The stable set of a period-$k$ point $x_0$ is defined as $W^s(x_0) := \{x : \lim_{n \to \infty} f^{(kn)}(x) = x_0.\}$.

---

[1]Analysis in Wang et al. (2022) and Chen and Bruna (2023) is also global, but not applicable to our model.

The stable set of the orbit of a periodic point $x_0$ is the union of the stable sets of all points in the orbit of $x_0$. A point $x_0$ is an aperiodic point if it is not an asymptotically periodic point and the orbit of $x_0$ is bounded. We say a fixed point $x_0$ of $f$ is stable if, for any $\epsilon > 0$, there is a $\delta > 0$ such that for any $x$ satisfiying $|x - x_0| < \delta$, we have $|f^{(n)}(x) - x_0| < \epsilon$ for all $n \geq 0$. The fixed point $x_0$ is said to be unstable if it is not stable. The fixed point $x_0$ is asymptotically stable if it is stable and there is an $\delta > 0$ such that $\lim_{n\to\infty} f^{(n)}(x) = x_0$ for all $x$ satisfying $|x - x_0| < \delta$. A period-$p$ point $x_0$ and its associated periodic orbit are asymptotically stable if $x_0$ is an asymptotically stable fixed point of $f^{(p)}$. A point $x_0 \in \mathbb{R} \bigcup \{+\infty, -\infty\} \backslash S$ is called an absorbing boundary point of $S$ for $f$ with period $p$, for some $p \in \{1, 2\}$, if there exists an open set $U \subseteq S$ such that $\lim_{k\to\infty} f^{(pk)}(y) \to x$ for all $y \in U$. The Schwarzian derivative[2] of a three-times continuously differentiable function $f$ is defined (at non-critical points) as

$$\mathsf{S}f(x) := (f'''(x)/f'(x)) - 1.5 \left(f''(x)/f'(x)\right)^2, \text{ where } f'(x) \neq 0.$$

The Lyapunov exponent[3] of a given orbit with initialization $x_0$ is defined as $\mathsf{L}f(x_0) = \lim_{n\to\infty} \frac{1}{n} \sum_{i=1}^{n-1} \log |f'(x_i)|$. The sharpness of a loss function is defined as the maximum eigenvalue of the Hessian matrix of the loss.

**Bifurcation analysis.** Our main goal in this section is to undertake a bifurcation analysis of the following discrete dynamics system defined by a cubic map. For $a > 0$, first define the functions $g$ and $f$, parameterized by $a$, as

$$g_a(z) = z^2 + (a - 2)z + 1 - 2a = (z + a)(z - 2) + 1 \quad \text{and} \quad f_a(z) = zg_a(z). \tag{2.1}$$

Next, consider the discrete dynamical system given by

$$z_{t+1} = f_a(z_t) = z_t g_a(z_t). \tag{2.2}$$

Note that for any $a, \epsilon > 0$ and $z_0 \geq 2 + \epsilon$ or $z_0 \leq -a - \epsilon$, we will have $\lim_{t\to\infty} |z_t| = +\infty$. Hence, we only study the case when $z_0 \in [-a, 2]$. We will show in Section 3 that the dynamics of the training loss for several quadratic regression models could be captured by (2.2). The parameter $a$ in (2.1) for the models will naturally correspond to the step-size of the gradient descent algorithm.

We next introduce the precise definitions of the five phase that arise in the bifurcation analysis of (2.1). To do so, we need the following definition of chaos in the Li-Yorke sense (Li and Yorke, 1975). Li-Yorke chaos is widely used in the study of dynamical systems and is also directly related to important measures of the complexity of dynamical systems, like the topological entropy (Adler et al., 1965; Franzová and Smítal, 1991). We also refer to Aulbach and Kieninger (2001) and Kolyada (2004) for its relationship to other notions of chaos and related history.

**Definition 1** (Li-Yorke Chaos (Li and Yorke, 1975))**.** Suppose we are given a function $f(x)$. If there exists a compact interval $I$ such that $f : I \to I$, then it is called Li-Yorke chaotic (Li and Yorke, 1975; Aulbach and Kieninger, 2001) when it satisfies

- For every $k = 1, 2, ...$ there is a periodic point in $I$ having period-$k$.
- There is an uncountable set $S \subseteq I$ (containing no periodic points), which satisfies for any $p, q \in S$ with $p \neq q$, $\limsup_{t\to\infty} |f^{(t)}(p) - f^{(t)}(q)| > 0$, $\liminf_{t\to\infty} |f^{(t)}(p) - f^{(t)}(q)| = 0$, and for any $p \in S$ and periodic point $q \in I$, $\limsup_{n\to\infty} |f^{(t)}(p) - f^{(t)}(q)| > 0$.

To define the 5 phases in particular, we consider the orbit $\{f^{(k)}(x)\}_{k=0}^{+\infty}$ generated by a given function $f$ defined over a set $I$, in which the initial point $x$ belongs to.

**Definition 2.** Given a function $f(x)$ defined on a set $I$, we say the discrete dynamics is in the

- **Monotonic phase**, when $\{|f^{(k)}(x)|\}_{k=0}^{\infty}$ is decreasing and $\lim_{n\to\infty} |f^{(n)}(x)| = 0$ for almost every $x \in I$.
- **Catapult phase**, when $\{|f^{(k)}(x)|\}_{k=m}^{\infty}$ is not decreasing for any $m$ and $\lim_{n\to\infty} |f^{(n)}(x)| = 0$ for almost every $x \in I$. We say such sequences have catapults.

---

[2]It is widely used in the study of dynamical systems for its sign-preservation property under compositions; see, for example, De Melo and Van Strien (2012).

[3]It is associated with the stability properties and commonly used in dynamical systems to measure the sensitive dependence on initial conditions (Strogatz, 2018).

- **Periodic phase**, when $f$ is not Li-Yorke chaotic, $\{|f^{(k)}(x)|\}_{k=0}^{\infty}$ is bounded and does not have a limit for almost every $x \in I$, and there exists period-2 points in $I$.
- **Chaotic phase**, when the function $f$ is Li-Yorke chaotic and $\{|f^{(k)}(x)|\}_{k=0}^{\infty}$ is bounded for almost every $x \in I$.
- **Divergent phase**[4], when $\lim_{n \to \infty} |f^{(n)}(x)| = +\infty$ for almost every $x \in I$.

As an illustration, in Figure 1, we plot the five phases for the parameterized function and its discrete dynamical system defined in (2.1) with initialization 1.9, i.e., $x_k = f_a^{(k)}(x_0)$, $x_0 = 1.9$. We have the following main result for different phases of dynamics.

**Theorem 2.1.** *Suppose $f_a(z)$ is defined in (2.1). Define $z_{t+1} = f_a(z_t)$ with $z_0$ sampled uniformly at random in $(-a, 2)$. Then there exists $a_* \in (1, 2)$ such that the following holds.*

- *If $a \in (0, 2\sqrt{2} - 2]$, then almost surely $\lim_{t \to \infty} |z_t| = 0$ and $|z_t|$ is decreasing, and hence the dynamics is in the monotonic phase.*
- *If $a \in (2\sqrt{2} - 2, 1]$, then almost surely $\lim_{t \to \infty} |z_t| = 0$ and $|z_t|$ have catapults, and hence the dynamics is in the catapult phase.*
- *If $a \in (1, a_*)$, then there exists a period-2 point in $(0, 1)$. $z_t \in (-a, 2)$ for all $t$. If there exists an asymptotically stable periodic orbit, then the orbit of $z_0$ is asymptotically periodic almost surely, and hence the dynamics is in the periodic phase.*
- *If $a \in (a_*, 2]$, $f_a$ is Li-Yorke chaotic. $z_t \in (-a, 2)$ for all $t$. If there exists an asymptotically stable periodic orbit, then the orbit of $z_0$ is asymptotically periodic almost surely, and hence the dynamics is in the chaotic phase.*
- *If $a \in (2, +\infty)$, then $\lim_{t \to \infty} |z_t| = +\infty$ almost surely, and hence the dynamics is in the divergent phase.*

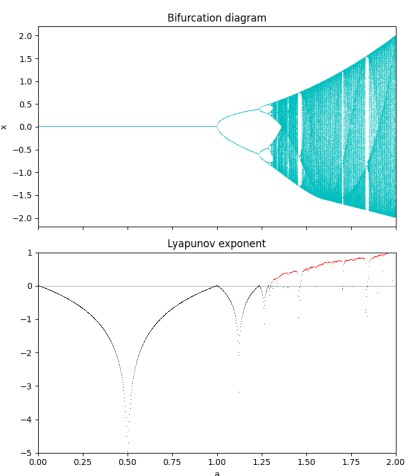

Figure 2: Bifurcation diagram and Lyapunov exponent. Initialization $z_0 = 0.1$.

In Figure 2 we numerically plot a bifurcation diagram for $a \in (0, 2)$ and Lyapunov exponent scatter plot with initialization $z_0 = 0.1$. The main ingredients in proving Theorem 2.1 are the following Lemmas 1, 2, and 3. Note that by straightforward computations, we have

$$f_a'(0) = 1 - 2a \in (-1, 1) \Leftrightarrow a \in (0, 1).$$

This implies 0 is a asymptotically stable fixed point when $a \in (0, 1)$. This type of local stability analysis is standard in dynamical systems literature (Hale and Koçak, 2012; Strogatz, 2018), and has been used in analyzing the training dynamics of gradient descent recently (Zhu et al., 2022; Song and Yun, 2023). However, such results are limited to only local regions. In contrast, the following results provide a global convergence analysis.

**Lemma 1.** *Suppose $0 < a \le 1$ and $-a \le z_0 \le 2$. Then we have*

- *(i) $-a \le z_t \le 2$ for any $t$, and $f_a$ does not have a period-2 point on $[-a, 2]$.*
- *(ii) If $z_0$ is chosen from $[-a, 2]$ uniformly at random,*

then $\lim_{t \to \infty} z_t = 0$ almost surely. Moreover, if $0 < a \le 2\sqrt{2} - 2$, then almost surely $|z_{t+1}| \le |z_t|$ for all $t$. If $2\sqrt{2} - 2 < a \le 2$, then almost surely $\{|z_t|\}_{t=0}^{\infty}$ has catpults.

**Lemma 2.** *Suppose $1 < a \le 2$ and $-a \le z_0 \le 2$. Then we have*

- *(i) $-a \le z_t \le 2$ for any $t$, and $f_a(z)$ has a period-2 point on $[0, 1]$.*
- *(ii) There exists $a_* \in (1, 2)$ such that for any $a \in (a_*, 2)$, $f_a$ is Li-Yorke chaotic, and for any $a \in (1, a_*)$, $f_a$ is not Li-Yorke chaotic.*
- *(iii) If there exists an asymptotically stable orbit and $z_0$ is chosen from $[-a, 2]$ uniformly at random, then the orbit of $z_0$ is asymptotically periodic almost surely.*

---

[4]We do not further sub-characterize the divergent phase as it is uninteresting.

**Lemma 3.** *Suppose $a > 2$. $z_0$ is chosen from $[-a, 2]$ uniformly at random. Then $\lim_{t \to \infty} |z_t| = +\infty$ almost surely.*

In Lemma 2, part (iii), we assume the existence of an asymptotically stable periodic point. Note that such a point must have negative Lyapunov exponent (Strogatz, 2018). It is possible to obtain particular values for $a$ under which $f_a(z)$ has an asymptotically stable orbit. For example, $a$ can be chosen such that $|f_a'(p) f_a'(q)| < 1$, where $p \in (0, 1)$ is a period-2 point with $f_a(p) = q$. In Figure 2 we plot the Lyapunov exponent of $f_a$ at the orbit starting from $z_0 = 0.1$. It is interesting to explicitly characterize the set of $a$ values in $(1, 2)$ such that $f_a(z)$ has an asymptotically stable periodic orbit. Furthermore, we conjecture that $a_*$ defined in Lemma 2 is the smallest number $a \in (1, 2)$ such that $(1 - 2a)/3$ is a period-3 point. The above two problems are challenging and left as future work.

## 3 APPLICATIONS TO QUADRATIC REGRESSION MODELS

We now provide illustrative examples based on quadratic or second-order regression models, motivated by the works of Zhu et al. (2022); Agarwala et al. (2023). Specifically, we consider a generalized phase retrieval model and training hidden-layers of 2-layer neural networks with quadratic activation function as examples.

### 3.1 EXAMPLE 1: GENERALIZED PHASE RETRIEVAL

**Single Data Point.** Following Zhu et al. (2022), it is instructive to study the dynamics with a single training sample. Consider the following optimization problem on a single data point $(X, y)$:

$$\min_w \left\{ \ell(w) = \frac{1}{2}(g(w; X) - y)^2 \right\}, \quad \text{where } g(w; X) = \frac{\gamma(X^\top w)^2}{2} + cX^\top w, \quad (3.1)$$

where $\gamma, c$ are arbitrary constants. The above model, with $\gamma = 2$ and, $c = 0$ corresponds to the classical phase retrieval model (also called as a single-index model with quadratic link function). We refer to Jaganathan et al. (2016) and Fannjiang and Strohmer (2020) for an overview, importance and applications of the phase retrieval model. We have the following Lemma for the training dynamics of gradient descent on solving (3.1).

**Theorem 3.1.** *Suppose we run gradient descent on (3.1) with step-size to be $\eta$. Define*

$$e^{(t)} := g(w^{(t)}; X) - y, \ z_t := \eta \gamma \|X\|^2 e^{(t)}, \ a = \left(\gamma y + \frac{c^2}{2}\right) \eta \|X\|^2. \quad (3.2)$$

*Then we have (i) $z_{t+1} = f_a(z_t)$ and thus Theorem 2.1 holds for $f_a$ and $z_t$; (ii) The sharpness is given by $\lambda_{max}(\nabla^2 \ell(w^{(t)})) = \frac{3z_t + 2a}{\eta}$.*

Note that Zhu et al. (2022) studied a related neural quadratic model (see their Eq. (3)). Here, we highlight that their results which does not cover our case. Indeed, defining $\eta_{\text{crit}} = 2/\lambda_{\max}(\nabla^2 \ell(w^{(0)}))$, according to their claim, catapults happen when $\eta_{\text{crit}} < \eta < 2\eta_{\text{crit}}$. In our notation, this condition is equivalent to $2 < 3z_0 + 2a < 4$. However this cannot happen because if the initialization $z_0$ is sufficiently small, say $z_0 = \mathcal{O}(\epsilon)$, then we know the previous condition become $1 - \mathcal{O}(\epsilon) < a < 2 - \mathcal{O}(\epsilon)$. However, according to Lemmas 1 and 2, we have that for $1 < a < 2$ the training dynamics is in the periodic or the chaotic phase and $z_t$ (and thus the loss function) will not converge to 0. Our theory (Lemma 1) suggests that catapults for quadratic regression model happens for almost every $z_0 \in (-a, 2)$ provided that $2\sqrt{2} - 2 < a \leq 1$. This intricate observation reveals that extending the current results on the catapult phenomenon from the model in Zhu et al. (2022) to our setting is not immediate and is actually highly non-trivial. We also notice that, interestingly, in the monotonic and catapult phases (i.e., $0 < a \leq 1$), we have the limiting sharpness satisfy $\lim_{t \to \infty} \lambda_{\max}(\nabla^2 \ell(w^{(t)})) = 2a/\eta = (2\gamma y + c^2) \|X\|^2$.

**Multiple Orthogonal Data Points.** We now consider gradient descent on quadratic regression on multiple data points that are mutually orthogonal. Suppose we are given a dataset $\{(X_i, y_i)\}_{i=1}^n$ with $\mathbf{X} = (X_1, ..., X_n)^\top$ satisfying $\mathbf{X}\mathbf{X}^\top = \text{diag}(\|X_1\|^2, ..., \|X_n\|^2)$. Consider the optimization problem defined by

$$\min_w \ell(w) := \frac{1}{n} \sum_{i=1}^n \ell_i(w) = \frac{1}{2n} \sum_{i=1}^n (g(w; X_i) - y_i)^2. \quad (3.3)$$

where $\ell_i(w)$ and $g(w; X_i)$ are as defined in (3.1).

**Theorem 3.2.** *Define the following:*

$$\alpha^{(t)}(X_i) := c(X_i) + \gamma X_i^\top w^{(t)}, \ \beta(X_i) := y_i + \frac{(c(X_i))^2}{2\gamma}, \ \kappa_n(X_i) := \frac{\eta\gamma \|X_i\|^2}{n},$$

$$e^{(t)}(X_i) := g(w^{(t)}; X_i) - y_i, \ z_i^{(t)} = \kappa_n(X_i)e^{(t)}(X_i), \ a_i = \beta(X_i)\kappa_n(X_i).$$

*If we run gradient descent on solving (3.3) with step-size $\eta$, then we have (i) $z_i^{(t+1)} = f_{a_i}(z_i^{(t)})$ and thus Theorem 2.1 hols for $f_{a_i}$ and $z_i^{(t)}$. (ii) The sharpness $\lambda_{\max}(\nabla^2\ell(w^{(t)})) = \max_{1\le i\le n}\frac{3z_i^{(t)} + 2a_i}{\eta}$.*

For this setup, the above theorem shows that the loss function is a summation of the loss on each individual data point. Recall that the training loss takes the form

$$\ell(w^{(t)}) = \frac{1}{2n}\sum_{i=1}^n \Big(g(w^{(t)}; X_i) - y_i\Big)^2 = \frac{1}{2n}\sum_{i=1}^n \frac{(z_i^{(t)})^2}{\kappa_n^2(X_i)} = \sum_{i=1}^n \frac{n(z_i^{(t)})^2}{2\eta^2\gamma^2 \|X_i\|^4}.$$

We can hence deduce that the dynamics is governed by $\max_{1\le i\le n} a_i$. In other words, for almost every $z^{(0)}$ in $\{z : -a_i \le z_i \le 2\}$, we have, by Lemma 1, that as long as $0 < a_i \le 1$ for all $i$, the training loss will converge to 0, and if $\max_{1\le i\le n} a_i > 1$, then by Lemma 2 we know that $\lim_{t\to\infty} |z_t| \ne 0$. We summarize this in the following corollary, which is a direct application of Theorems 2.1 and 3.2

**Corollary 1.** *Under the setup in Theorem 3.2, for almost all $z^{(0)} \in \{z : -a_i \le z_i \le 2\}$ we have*

- *If $0 < \max_{1\le i\le n} a_i \le 1$, then $\lim_{t\to\infty}\ell(w^{(t)}) = 0$. Moreover, if $0 < \max_{1\le i\le n} a_i \le 2\sqrt{2} - 2$, the sequence $\{\ell(w^{(t)})\}_{t=0}^\infty$ is decreasing.*
- *If $1 < \max_{1\le i\le n} a_i \le 2$, then $\{\ell(w^{(t)})\}_{t=0}^\infty$ is bounded and does not converge to 0.*
- *If $\max_{1\le i\le n} a_i > 2$, then $\lim_{t\to\infty}\ell(w^{(t)}) = +\infty$.*

## 3.2 EXAMPLE 2: NEURAL NETWORK WITH QUADRATIC ACTIVATION

In this section, we consider the following two layer neural networks with its loss function on data point $(X_i, y_i)$ defined as:

$$g(u, v; X_i) = \frac{1}{\sqrt{m}}\sum_{j=1}^m v_j\sigma\Big(\frac{1}{\sqrt{d}}u_j^\top X_i\Big), \ \ell_i = \frac{1}{2}\big(g(u, v; X_i) - y_i\big)^2$$

where the hidden-layer weights $u_i \in \mathbb{R}^d$ are to be trained and outer-layer weights $v_i \in \mathbb{R}$ are held constant, which corresponds to the feature-learning setting for neural networks. Also $m$ is the width of the hidden layer and $\sigma$ is the activation function. Define $\mathbf{U} := (u_1, ..., u_m)$. When the activation function is quadratic and $v_i = 1$ for all $i$, the loss function becomes

$$\min_{\mathbf{U}} \ell(\mathbf{U}) := \frac{1}{n}\sum_{j=1}^n \ell_j(\mathbf{U}) = \frac{1}{2n}\sum_{j=1}^n \Big(\frac{1}{\sqrt{md}}\sum_{i=1}^m (X_j^\top u_i)^2 - y_j\Big)^2. \tag{3.4}$$

As in the previous example, we assume $\mathbf{X}\mathbf{X}^\top = \mathrm{diag}(\|X_1\|^2, ..., \|X_n\|^2)$. We then have the following result on the gradient descent dynamics of the above problem.

**Theorem 3.3.** *Define the following:*

$$e_i^{(t)} = \frac{1}{\sqrt{md}}\sum_{j=1}^m (X_i^\top u_j^{(t)})^2 - y_i, \ z_i^{(t)} = \frac{2\eta \|X_i\|^2 e_i^{(t)}}{\sqrt{md}n}, \ a_i = \frac{2\eta \|X_i\|^2 y_i}{\sqrt{md}n}$$

*If we run gradient descent on solving problem (3.4) with step-size $\eta$, we have $z_i^{(t+1)} = f_{a_i}(z_i^{(t)})$ and thus Theorem 2.1 and Corollary 1 hold for $\ell(\mathbf{U}^{(t)})$.*

The orthogonal assumption that $\mathbf{X}\mathbf{X}^\top = \mathrm{diag}(\|X_1\|^2, ..., \|X_n\|^2)$, helps decouple the loss function across the samples and makes the evolution of the overall loss non-interacting (across the training samples). In order to relax this assumption, it is required to analyze bifurcation analysis of interacting dynamical systems, which is extremely challenging and not well-explored (Xu et al., 2021). In Section C.2, we present empirical results showing that similar phases exists in the general non-orthogonal setting as well. Theoretically characterizing this is left as an open problem.

## 4 EXPERIMENTAL INVESTIGATIONS

### 4.1 GRADIENT DESCENT DYNAMICS WITH ORTHOGONAL DATA FOR MODEL (3.4)

**Experimental setup.** We now conduct experiments to evaluate the developed theory. We consider gradient descent for training the hidden layers of a two-layer neural network with orthogonal training data, described in Section 3.2. Recall that $d, m$, and $n$ represents the dimension, hidden-layer width, and number of data points respectively. We set $d = 100, m \in \{5, 10, 25\}, n = 80$. We generate the ground-truth matrix $\mathbf{U}^* \in \mathbb{R}^{d \times m}$ where each entry is sampled from the standard normal distribution. The training data points collected in the data matrix, denoted as $\mathbf{X} \in \mathbb{R}^{n \times d}$, are the first $n$ rows of a randomly generated orthogonal matrix. The labels are generated via the model in Section 3.2, i.e., $y_i = \frac{1}{\sqrt{md}} \sum_{j=1}^{m} \left( X_i^\top u_j \right)^2 + \varepsilon_i$ where $\varepsilon_i$ is scalar noise sampled from a zero-mean normal distribution, with variances equal to $0, 0.25, 1$ in different experiments.

We set the step-size $\eta$ such that $\max_{1 \le i \le n} a_i$ defined in Theorem 3.2 belongs to the intervals of the first four phases. In particular, we choose $0.3, 0.9, 1, 1.2, 1.8$ for $m = 5, 10$ and $0.3, 0.9, 1, 1.2, 1.6$ for $m = 25$ (for each $m$, $0.9$ and $1$ are both in the catapult phase, and we pick $1$ since it is the largest step-size choice allowed in the catapult phase). The numbers $0, 1, 2, 3, 4$ of the plot labels correspond to these step-size choices respectively. In Figure 4 we present the training loss curves in log scale and the sharpness curves for $m = 25$. The horizontal axes denote the number of steps of gradient descent. In Section C.1, we also provide additional simulation results for different hidden-layer widths. From the training loss curves (left column) and the sharpness curves (middle column) we can clearly observe the four phases[5] thereby confirming our theoretical results.

### 4.2 PREDICTION BASED ON ERGODIC TRAJECTORY AVERAGING

A main take-away from our analysis and experiments so far is that gradient descent with large step-size behaves like *stochastic* gradient descent, except the randomness here is with respect to the orbit it converges to (in the non-monotonic phases). Recall that this viewpoint is also put-forward is several works, in particular Kong and Tao (2020). Hence, a natural approach is to do perform ergodic trajectory averaging to reduce the fluctuations (see right column in Figure 4). For any give point $X \in \mathbb{R}^d$, and any training iteration count $t$, the prediction $\hat{y}$ for the point $X$ is given by $\hat{y} := \frac{1}{t} \sum_{i=1}^{t} g(w^{(i)}; X)$, where $w^{(i)}$ corresponds to the training trajectory of the gradient descent algorithm trained with step-size $\eta$. Another way to think about the above prediction strategy is that the ergodic average approximates, in the limit, expectation with respect to the invariant distribution (supported on the orbit to which the trajectory converges to). A disadvantage of the ergodic averaging based prediction strategy described above is the test-time computational cost increases by $\mathcal{O}(t)$, per test point.

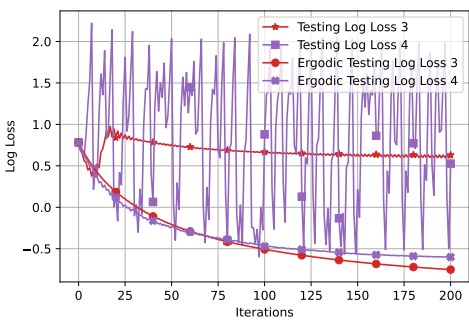

Figure 3: Test loss with and without averaging.

Figure 3 plots the testing loss for the model in (3.4), when trained with two values of large step-sizes ($\eta = 48, 60$). We observe from the figure that the ergodic trajectory averaging prediction smoothens the more chaotic testing loss. However, we also remark that from the plots in Figure 10[6], operating with slightly smaller step-size choice ($\eta = 36$) achieves the best testing error curves. See Section C.2 for additional observations. In the literature, ways of artificially inducing *controlled chaos* in the gradient descent trajectory has been proposed to obtain improved testing accuracy; see, for example, Lim et al. (2022). We believe the ergodic trajectory averaging based prediction methodology discussed above may prove to be fruitful to stabilize the testing loss in such cases as well. A detailed investigation of provable benefits of the ergodic trajector averaging predictor, is beyond the scope of the current work, and we leave it as intriguing future work.

---

[5]Here, we do not plot the divergent phase here for simplicity.

[6]Figure 10 provides a detailed comparison across various step-sizes, for different noise variances.

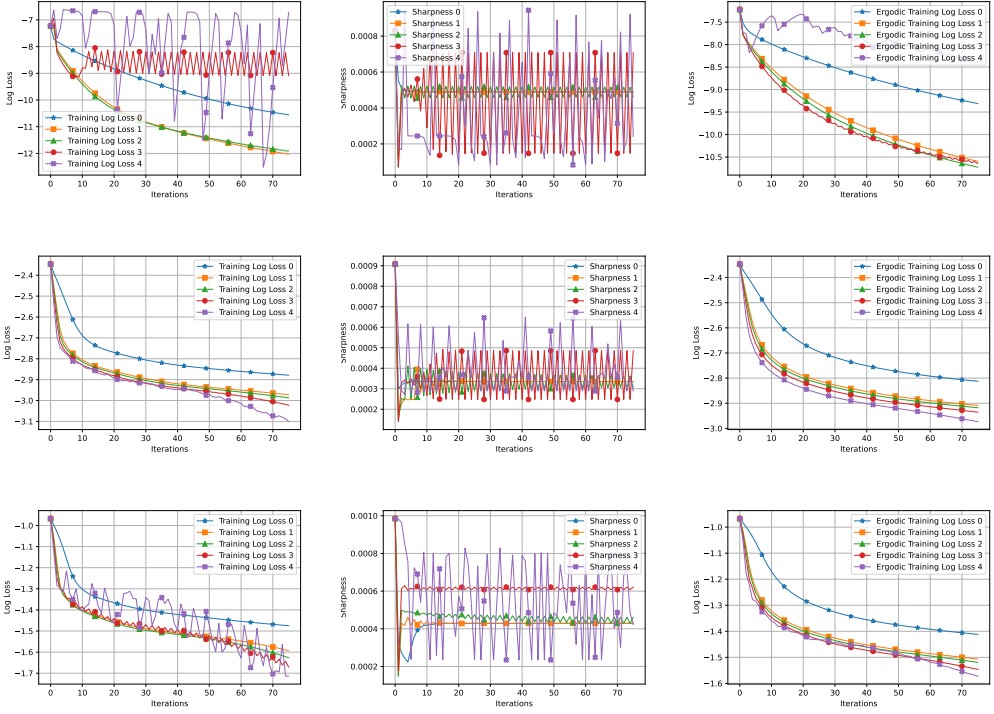

Figure 4: Hidden layer width = 25, with orthogonal data points. Rows from top to bottom represent different levels of noise – mean-zero normal distribution with variance $0, 0.25, 1$ respectively. The vertical axes are in log scale for the training loss curves. The second column is about the sharpness of the training loss functions.

### 4.3 Additional Experiments

Due to space limitation, we provide the following additional simulation results in the appendix:

- Section C.2 corresponds to non-orthogonal training data. We also include testing loss plots.
- Section C.3 corresponds to training the hidden-layer weights of a two-layer neural network with ReLU activation functions and non-orthogonal inputs.

## 5 Conclusion

Unstable and chaotic behavior is frequently observed when training deep neural networks with large-order step-sizes. Motivated by this, we presented a fine-grained theoretical analysis of a cubic-map based dynamical system. We show that the gradient descent dynamics is fully captured by this dynamical system, when training the hidden layers of a two-layer neural networks with quadratic activation functions with orthogonal training data. Our analysis shows that for this class of models, as the step-size of the gradient descent increases, the gradient descent trajectory has five distinct phases (from being monotonic to chaotic and eventually divergent). We also provide empirical evidence that show similar behavior occurs for generic non-orthogonal data. We empirically examine the impact of training in the different phases, on the generalization error, and observe that training in the phases of periodicity and chaos provides the highest test accuracy.

Immediate future works include: (i) developing a theoretical characterization of the training dynamics with generic non-orthogonal training data, which involves undertaking non-trivial bifurcation analysis of interacting dynamical systems, (ii) moving beyond quadratic activation functions and two-layer neural networks, and (iii) developing tight generalization bounds when training with large-order step-sizes. Overall, our contributions make concrete steps towards developing a fine-grained understanding of the gradient descent dynamics when training neural networks with iterative first-order optimization algorithms with large step-sizes.

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

# A   PROOFS OF MAIN RESULTS

## A.1   PROOFS OF RESULTS IN SECTION 2

We first present several technical results required to prove our main results.

**Lemma 4.** *Let $f(x)$ be a polynomial. If all the roots of $f'(x)$ are real and distinct, then we have*

$$\mathsf{S}f(x) = \frac{f'''(x)}{f'(x)} - \frac{3}{2}\left(\frac{f''(x)}{f'(x)}\right)^2 < 0 \text{ for all } x \in I \text{ with } f'(x) \neq 0.$$

*Proof.* See, e.g., the proof of Proposition 11.2 in Devaney (1989). ∎

**Lemma 5.** *Suppose we are given a real-valued continuous function $f(x) : \mathbb{R} \to \mathbb{R}$ and a bounded closed interval $I \subseteq \mathbb{R}$ with $x_0 \in I$. Define $x_k := f^{(k)}(x_0)$. If the sequence $\{x_k\}_{k=0}^{\infty}$ is monotonic, then one of the following holds.*

- *(i) $\{x_k\}_{k=0}^{\infty} \subsetneq I$, i.e., there exists $x_t \notin I$ for some t.*

- *(ii) $\{x_k\}_{k=0}^{\infty} \subseteq I$, and $\lim_{t \to \infty} f^{(t)}(x_0)$ exists and is a fixed point of $f(x)$ in $I$.*

*Proof.* If (i) holds, then the conclusion is true. When (i) does not hold, then $\{x_k\}_{k=0}^{\infty} \subseteq I$. Since this sequence is monotonic and included in a bounded closed interval, we know its limit exists and is in $I$. Moreover, we have

$$\lim_{t \to \infty} x_t = \lim_{t \to \infty} x_{t+1} = \lim_{t \to \infty} f(x_t) = f(\lim_{t \to \infty} x_t),$$

where the last equality holds since $f$ is continuous. Clearly $\lim_{t \to \infty} x_t$ is a fixed point of $f$. ∎

The following lemma characterizes the basic properties of the cubic function $f_a$ defined in (2.1).

**Lemma 6.** *Suppose $a > 0$. Then $f_a(z)$ has the following properties.*

- *(i) The local minimum and maximum of $f_a(z)$ are at $z = 1$ and $z = \frac{1-2a}{3}$ respectively, and*

$$f_a(1) = -a, \ f_a\left(\frac{1-2a}{3}\right) = \frac{(2a-1)(2a^2+7a-4)}{27} = \frac{4a^3+12a^2-15a+4}{27}.$$

- *(ii) $f_a(z)$ is monotonically increasing on $[-a, \frac{1-2a}{3}]$, monotonically decreasing on $[\frac{1-2a}{3}, 1]$, and monotonically increasing on $[1, 2]$.*

- *(iii) For any $-a \le z \le 2$, we have $-a \le f_a(z) \le \max\left\{f_a\left(\frac{1-2a}{3}\right), 2\right\}$. Moreover, $f_a\left(\frac{1-2a}{3}\right) \le 2$ if and only if $a \le 2$.*

*Proof.* Note that we have

$$f_a'(z) = 3z^2 + 2(a-2)z + (1-2a) = (z-1)(3z+2a-1). \tag{A.1}$$

which implies 1 and $\frac{1-2a}{3}$ are critical points of $f_a(z)$. Moreover, by $f_a''(z) = 6z + 2a - 4$ we know $f_a''(1) > 0$ and $f_a''(\frac{1-2a}{3}) < 0$. Hence, they are local minimum and maximum respectively. The rest of (i) is true by calculation. (ii) is true by noticing the expression of $f_a'(z)$ in (A.1). (iii) is a direct conclusion of (i) and (ii) since for $-a \le z \le 2$ we have

$$-a = \min\{f_a(1), f_a(-a)\} \le f_a(z) \le \max\left\{f_a\left(\frac{1-2a}{3}\right), f_a(2)\right\}.$$

By (i) and some calculation we know

$$f_a\left(\frac{1-2a}{3}\right) - 2 = \frac{4a^3+12a^2-15a-50}{27} = \frac{(2a+5)^2(a-2)}{27}.$$

This proves the rest of (iii). ∎

**Lemma 7.** *Suppose $2\sqrt{2} - 2 < a \le 1$. Define five subintervals of $[-a, 2]$ as follows.*

$$I_1 = \left[ -a, \frac{2 - a - \sqrt{a^2 + 4a}}{2} \right], \ I_2 = \left[ \frac{2 - a - \sqrt{a^2 + 4a}}{2}, 0 \right],$$

$$I_3 = [0, 0.25], \ I_4 = \left[ 0.25, \frac{2 - a + \sqrt{a^2 + 4a}}{2} \right], \ I_5 = \left[ \frac{2 - a + \sqrt{a^2 + 4a}}{2}, 2 \right].$$

*Then we have*

- *(i) $f_a(I_1) \subseteq I_1 = I_2$, $f_a(I_4) = I_1 \cup I_2$, $f_a(I_5) = I_3 \cup I_4 \cup I_5$.*
- *(ii) $f_a(I_2) \subseteq I_3$, $f_a(I_3) \subseteq I_2$.*

*Proof.* We first prove (i). By Lemma 6 we know $f_a(z)$ is increasing on $I_1$, achieving its local minimum at $z = 1$ on $I_4$, increasing on $I_5$, then we know

$$f_a(I_1) = \left[ f_a(-a), f_a\left( \frac{2 - a - \sqrt{a^2 + 4a}}{2} \right) \right] = [-a, 0] = I_1 \cup I_2.$$

$$f_a(I_4) = \left[ f_a(1), \max\left\{ f_a(0.25), f_a\left( \frac{2 - a + \sqrt{a^2 + 4a}}{2} \right) \right\} \right] = [-a, 0] = I_1 \cup I_2.$$

$$f_a(I_5) = \left[ f_a\left( \frac{2 - a + \sqrt{a^2 + 4a}}{2} \right), f_a(2) \right] = [0, 2] = I_3 \cup I_4 \cup I_5.$$

This completes the proof of (i).

To prove (ii), observe that when $a \in (2\sqrt{2} - 2, 1]$ we have $\frac{2 - a - \sqrt{a^2 + 4a}}{2} < \frac{1 - 2a}{3} < 0$. By Lemma 6 we know the local maximum of $f_a$ over $I_2 = \left[ \frac{2 - a - \sqrt{a^2 + 4a}}{2}, 0 \right]$ is achieved at $\frac{1 - 2a}{3}$, this together with the fact that $f_a(0) = f_a\left( \frac{2 - a - \sqrt{a^2 + 4a}}{2} \right) = 0$ implies

$$f_a(I_2) = \left[ f_a(0), f_a\left( \frac{1 - 2a}{3} \right) \right] = \left[ 0, \frac{4a^3 + 12a^2 - 15a + 4}{27} \right] \subseteq [0, 0.25],$$

where the last subset inclusion is true since

$$(4a^3 + 12a^2 - 15a + 4)' = 12a^2 + 24a - 15 > 0, \ \forall a \in (2\sqrt{2} - 2, 1].$$

This implies when $a \in (2\sqrt{2} - 2, 1]$,

$$\frac{4a^3 + 12a^2 - 15a + 4}{27} \le \frac{(4a^3 + 12a^2 - 15a + 4)|_{a=1}}{27} = \frac{5}{27} < 0.25.$$

On the other hand, we know from Lemma 6 that on $I_3 = [0, 0.25] (\subseteq \left[ \frac{1 - 2a}{3}, 1 \right])$ $f_a$ is decreasing. Hence,

$$f_a(I_3) = [f_a(0.25), f_a(0)] = \left[ -\frac{7}{16}a + \frac{9}{16}, 0 \right] \subseteq \left[ \frac{2 - a - \sqrt{a^2 + 4a}}{2}, 0 \right] = I_2.$$

where the last subset inclusion is true since

$$f_a(0.25) = -\frac{7}{16}a + \frac{9}{16} > \frac{2 - a - \sqrt{a^2 + 4a}}{2}, \ \forall a \in (2\sqrt{2} - 2, 1].$$

This completes the proof of (ii). ∎

See Figure 5(a) for a visualization of the subintervals $I_1, ..., I_5$ for $a = 1$ and an example of the orbit on it.

**Lemma 8.** *Suppose $0 < a \le 1$ and $-a \le z_0 \le 2$. Then we have*

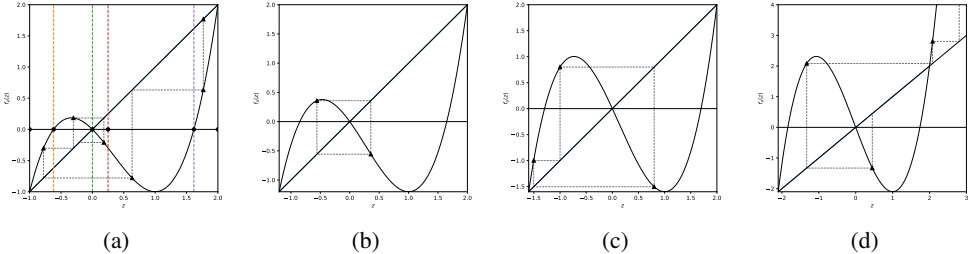

Figure 5: From left to right: cubic function $f_1(z)$ with different regions diveded by subintervals and a trajectory of $\{z_i\}_{i=0}^5$, cubic function $f_{1.2}(z)$ with two period-2 point, cubic function $f_{1.6}(z)$ with a period-3 point, and cubic function $f_{2.1}(z)$ with a diverging orbit. We have the cubic curve and the identical mapping line as the solid curves. We use four colored dashed lines in Figure 5(a) to represent the boundaries that are orthogonal to the endpoints of $I_2$ and $I_4$ defined in Lemma 7 respectively. The triangle markers represent some terms of a certain orbit, in which horizontal and vertical dotted lines visualize the transitioning trajectory between consecutive terms in an orbit.

- *(i) $-a \leq z_t \leq 2$ for any t, and $f_a$ does not have a period-2 point on $[-a, 2]$.*

- *(ii) If $z_0$ is chosen from $[-a, 2]$ uniformly at random, then $\lim_{t \to \infty} z_t = 0$ almost surely. Moreover, if $0 < a \leq 2\sqrt{2} - 2$, then almost surely $|z_{t+1}| \leq |z_t|$ for all t. If $2\sqrt{2} - 2 < a \leq 2$, then almost surely $\{|z_t|\}_{t=0}^\infty$ has catpults.*

*Proof.* The boundedness of each iterate (i.e., $z_t \in [-a, 2]$) can be proved by using simple induction and Lemma 6, $0 < a \leq 1$, and $-a \leq z_0 \leq 2$. To prove the rest of (i), by (2.1) we know a period-2 point is a solution of

$$f_a^{(2)}(z) = z, \; f_a(z) \neq z$$

which are equivalent to

$$g_a(z)g_a(zg_a(z)) = 1, z \notin \{-a, 0, 2\}. \tag{A.2}$$

Hence it suffices to prove (A.2) do not have a solution. Define

$$h_a(z) = g_a(z) - 1 = (z + a)(z - 2) < 0, \; \forall z \in (-a, 2).$$

We have

$$
\begin{aligned}
&g_a(z)g_a(zg_a(z)) - 1 \\
=&h_a(z) + h_a(z)h_a(zg_a(z)) + h_a(zg_a(z)) \\
=&h_a(z)(1 + h_a(zg_a(z))) + (z + a + zh_a(z))(z - 2 + zh_a(z)) \\
=&h_a(z)(1 + h_a(zg_a(z))) + h_a(z) + (z(z - 2) + z(z + a))h_a(z) + z^2 h_a^2(z) \\
=&h_a(z)(h_a(zg_a(z)) + z^2 h_a(z) + 2z^2 + (a - 2)z + 2). \tag{A.3}
\end{aligned}
$$

We have

$$
\begin{aligned}
&h_a(zg_a(z)) + z^2 h_a(z) + 2z^2 + (a - 2)z + 2 \\
=&(zg_a(z) + a)(zg_a(z) - 2) + z^2(z + a)(z - 2) + 2z^2 + (a - 2)z + 2 \\
=&z^2(z^2 + (a - 2)z + 1 - 2a)^2 + (a - 2)z(z^2 + (a - 2)z + 1 - 2a) - 2a \\
&+ z^2(z + a)(z - 2) + 2z^2 + (a - 2)z + 2 \\
=&z^6 + (2a - 4)z^5 + (a^2 - 8a + 7)z^4 - (4a^2 - 12a + 8)z^3 + (5a^2 - 10a + 7)z^2 \\
&- (2a^2 - 6a + 4)z + 2 - 2a \\
=&(z^2 + (a - 1)z + 1 - a)(z^4 + (a - 3)z^3 + (3 - 3a)z^2 + (2a - 2)z + 2). \tag{A.4}
\end{aligned}
$$

Observe that

$$z^2 + (a-1)z + (1-a) \geq (1-a) - \frac{(a-1)^2}{4} = \frac{(3+a)(1-a)}{4} \geq 0, \ \forall a \in (0,1]. \quad \text{(A.5)}$$

The equalities hold if and only if $z = 0$, $a = 1$. We also have

$$z^4 + (a-3)z^3 + (3-3a)z^2 + (2a-2)z + 2 > 0, \ \forall z \in \{0, 1, 2\}$$
$$z^4 + (a-3)z^3 + (3-3a)z^2 + (2a-2)z + 2$$
$$= z(z-1)(z-2)\left(a + z + \frac{1}{z} + \frac{1}{z^2 - 3z + 2}\right), \forall z \notin \{0, 1, 2\}.$$

For different $z$ we can verify the following inequalities via basic algebra or Young's inequality:

$$z(z-1)(z-2) < 0, \ \left(a + z + \frac{1}{z} + \frac{1}{z^2 - 3z + 2}\right) < 1 + 2 + \frac{1}{2} + \frac{1}{-0.25} < 0, \qquad \forall z \in (1, 2).$$

$$z(z-1)(z-2) > 0, \ \left(a + z + \frac{1}{z} + \frac{1}{z^2 - 3z + 2}\right) > 0 + 1 + 1 + 0 > 0, \qquad \forall z \in (0, 1).$$

$$z(z-1)(z-2) < 0, \ \left(a + z + \frac{1}{z} + \frac{1}{z^2 - 3z + 2}\right) < 1 - 1 - 1 + \frac{1}{2} < 0, \qquad \forall z \in (-a, 0).$$

Thus we may conclude that

$$z^4 + (a-3)z^3 + (3-3a)z^2 + (2a-2)z + 2 > 0, \ \forall z \in (-a, 2). \quad \text{(A.6)}$$

By (A.3), (A.4), (A.5), (A.6), we know $g_a(z)g_a(zg_a(z)) - 1 \neq 0$ if $z \notin \{-a, 0, 2\}$. Hence $f_a$ does not have a period-2 point on $[-a, 2]$.

To prove the first part in (ii) (the limit converges to 0 almost surely), we will prove

$$(1) \ \lim_{t \to \infty} z_t \in \{-a, 0, 2\}, \ (2) \ \text{The set } S \text{ such that the orbit with } z_0 \in S \text{ has measure } 0. \quad \text{(A.7)}$$

We now consider two cases – $a \in (0, 2\sqrt{2} - 2]$ and $a \in (2\sqrt{2} - 2, 1]$.

**Case 1:** $a \in (0, 2\sqrt{2} - 2]$**.** Note that we have

$$|g_a(z_t)| = |z_t^2 + (a-2)z_t + 1 - 2a| \leq \max\left(|g_a(-a)|, |g_a(2)|, |g_a\left(1 - \frac{a}{2}\right)|\right) = 1,$$

where the last equality holds since $g_a(-a) = g_a(2) = 1$ and $|g_a\left(1 - \frac{a}{2}\right)| = \frac{a^2 + 4a}{4} \leq 1$ for any $a \in (0, 2\sqrt{2} - 2]$. Hence, we know

$$|z_{t+1}| = |f_a(z_t)| = |z_t g_a(z_t)| \leq |z_t|, \ \forall z_t \in [-a, 2] \quad \text{(A.8)}$$

Hence $\lim_{t \to \infty} |z_t|$ exists.

$$\lim_{t \to \infty} |z_t| = \lim_{t \to \infty} |z_{t+1}| = \lim_{t \to \infty} |z_t| |g_a(z_t)|$$

Hence, we know

$$\lim_{t \to \infty} |z_t| = 0, \ \text{or} \ \lim_{t \to \infty} |z_t| \neq 0, \ \lim_{t \to \infty} |g_a(z_t)| = 1.$$

If $\lim_{t \to \infty} |z_t| \neq 0$, then we have two subcases

- Sub-case 1: $\lim_{t \to \infty} z_t$ exists. We can verify that

$$\lim_{t \to \infty} z_t = \lim_{t \to \infty} z_{t+1} = f_a(\lim_{t \to \infty} z_t)$$

and thus $\lim_{t \to \infty} z_t$ is one of the fixed points of $f_a(z) \in \{-a, 0, 2\}$.

- Sub-case 2: $\lim_{t\to\infty} z_t$ does not exist. Since $\lim_{t\to\infty} |z_t|$ exists, we know there exists an infinite subsequence (denoted as $A_1$) of $\{z_t\}_{t=0}^{\infty}$ with some limit $c$ and the complement of the sequence, as another infinite subsequence (denoted as $A_2$), has limit $-c$ for some constant $c > 0$. Hence, we can pick a sequence of the subscripts $k_1 < k_2 < ... < k_n < ...$ such that $z_{k_1}, ..., z_{k_n}, ...$ belong to $A_1$ and $z_{k_1+1}, ..., z_{k_n+1}, ...$ belong to $A_2$. Moreover, we have

$$c = \lim_{i\to\infty} z_{k_i} = -\lim_{i\to\infty} z_{k_i+1} = -\lim_{i\to\infty} z_{k_i} g_a(z_{k_i}) = -c g_a(c)$$

This implies that $g_a(c) = -1$, i.e.,

$$c^2 + (a-2)c + 2 - 2a = 0.$$

From its discriminant $(a-2)^2 - 4(2-2a) = a^2 + 4a - 4 \leq 0$ for $a \in (0, 2\sqrt{2}-2]$ where equality holds only at $2\sqrt{2} - 2$, we know $a = 2\sqrt{2} - 2$ and thus $c = 2 - \sqrt{2}$. However, we can apply the similar trick and pick another sequence $\tilde{k}_1 < \tilde{k}_2 < ... < \tilde{k}_n < ...$ such that $z_{\tilde{k}_1}, ..., z_{\tilde{k}_n}, ...$ belong to $A_2$ and $z_{\tilde{k}_1+1}, ..., z_{\tilde{k}_n+1}, ...$ belong to $A_1$. This implies

$$-c = \lim_{i\to\infty} z_{\tilde{k}_i} = -\lim_{i\to\infty} z_{\tilde{k}_i+1} = -\lim_{i\to\infty} z_{k_i} g_a(z_{k_i}) = -(-c) g_a(-c)$$

which gives

$$c^2 - (a-2)c + 2 - 2a = 0.$$

This contradicts with $a = 2\sqrt{2} - 2$ and $c = 2 - \sqrt{2}$. This means case 2 does not exist.

Hence, we know $|z_t|$ is decreasing (not necessarily strictly) and $\lim_{t\to\infty} z_t \in \{-a, 0, 2\}$.

**Case 2:** $a \in (2\sqrt{2} - 2, 1]$. We divide the interval $[-a, 2]$ into the following five parts:

$$I_1 = \left[-a, \frac{2-a-\sqrt{a^2+4a}}{2}\right], \ I_2 = \left[\frac{2-a-\sqrt{a^2+4a}}{2}, 0\right],$$

$$I_3 = [0, 0.25], \ I_4 = \left[0.25, \frac{2-a+\sqrt{a^2+4a}}{2}\right], \ I_5 = \left[\frac{2-a+\sqrt{a^2+4a}}{2}, 2\right].$$

Recall that by Lemma 7 we have:

$$f_a(I_1) = I_1 \cup I_2, \ f_a(I_2) \subseteq I_3, \ f_a(I_3) \subseteq I_2, \ f_a(I_4) = I_1 \cup I_2, \ f_a(I_5) = I_3 \cup I_4 \cup I_5.$$

We have the following conclusion. Observe that $f_a$ is continuous, and

$$z_{t+1} - z_t = f_a(z_t) - z_t = z_t(z_t + a)(z_t - 2) \geq 0, \ \forall z_t \in I_1 = \left[-a, \frac{2-a-\sqrt{a^2+4a}}{2}\right],$$

$$z_{t+1} - z_t = f_a(z_t) - z_t = z_t(z_t + a)(z_t - 2) \leq 0, \ \forall z_t \in I_5 = \left[\frac{2-a+\sqrt{a^2+4a}}{2}, 2\right].$$

We know if the sequence $\{z_t\}_{t=0}^{\infty}$ visits $I_5$, by Lemma 5 we know either $\lim_{t\to\infty} z_t = 2$ or there exists $M > 0$ such that $z_t \notin I_5$ for any $t \geq M$. Then if the sequence visits $I_1$ then by Lemma 5 either $\lim_{t\to\infty} z_t = -a$ or there exists $\tilde{M} > M > 0$ such that $z_t \in I_2 \cup I_3$ for any $t \geq \tilde{M}$, since $f_a(I_1) \subseteq I_1 \cup I_2$ and $f_a(I_2 \cup I_3) \subseteq I_2 \cup I_3$. Hence, the proof is reduced to the case when $z_0 \in I_2 \cup I_3$. For the case when $z_0 \in I_2 \cup I_3 = \left[\frac{2-a-\sqrt{a^2+4a}}{2}, 0.25\right]$. The key observation is to show that in this interval

$$|z_{t+2}| \leq |z_t|. \tag{A.9}$$

Recall that by Lemma 7 (ii) we have

$$f_a(I_2) \subseteq I_3, \ f_a(I_3) \subseteq I_2. \tag{A.10}$$

To prove (A.9), we know it holds when $z_t = 0$. When $z_t \neq 0$, by (A.10) we know $f_a^{(2)}(z_t)$ and $z_t$ have the same sign provided $z_t \in I_2 \cup I_3 = \left[ \frac{2-a-\sqrt{a^2+4a}}{2}, 0.25 \right]$. This together with

$$f_a^{(2)}(z) = f_a(z)g_a(f_a(z)) = zg_a(z)g_a(zg_a(z))$$

implies that $g_a(z)g_a(zg_a(z)) \geq 0$ when $z \in \left[ \frac{2-a-\sqrt{a^2+4a}}{2}, 0 \right) \cup (0, 0.25]$. Thus we know

$$|z_{t+2}| = |z_t g_a(z_t)g_a(z_t g_a(z_t))| = |z_t||g_a(z_t)g_a(z_t g_a(z_t))|.$$

Thus to prove (A.9) it suffices to show $g_a(z)g_a(zg_a(z)) - 1 \leq 0$, which is true by combining (A.3), (A.4), (A.5), and (A.6). This completes the proof of (1) in (A.7). To prove (2) in (A.7), we first notice that $f_a(z) - z = z(z+a)(z-2) > 0$ for any $z \in (-a, 0)$, and thus $z_{t+1} > z_t$ for any $z_t$ near $-a$. Hence, $\lim_{t\to\infty} z_t = -a$ if and only if there exists $t$ such that $z_t = -a$. This implies that $f_a^{(t)}(z_0) = -a$ for some $t$. Similarly, $f_a(z) - z < 0$ for any $z \in (0, 2)$, which implies $z_{t+1} < z_t$ for any $z_t$ near 2. Hence, $\lim_{t\to\infty} z_t = 2$ if and only if $z_0 = 2$. Define

$$S = \bigcup_{n=0}^{\infty} f_a^{(-n)}(-a) \cup \{2\}$$

where $f_a^{(-n)}(-a)$ denotes the preimage of $-a$ under $f_a^{(n)}$. Clearly, each preimage is a finite set, and thus $S$ is countable. Hence, we know as long as $z_0 \in [-a, 2] \backslash S$, we have $\lim_{t\to\infty} z_t = 0$. Since $S$ is a countable set and $z_0$ is chosen uniformly at random, we know $\lim_{t\to\infty} z_t = 0$ almost surely.

For the rest of (ii), we have already proved in (A.8) that $\{|z_t|\}_{t=0}^{\infty}$ is decreasing when $0 < a \leq 2\sqrt{2} - 2$. To see $\{|z_t|\}_{t=0}^{\infty}$ has catapults when $2\sqrt{2} - 2 < a \leq 1$, we consider the following intervals

$$J_1 = [-a, 0] = I_1 \cup I_2, \ J_2 = \left[ 0, \min \left\{ \frac{2 - a + \sqrt{a^2 + 4a - 4}}{2}, 0.25 \right\} \right] \subseteq I_3,$$

where we have $a^2 + 4a - 4 > 0$ for $a > 2\sqrt{2} - 2$ so $J_2$ is well-defined. Notice that

$$0 < z < \frac{2 - a + \sqrt{a^2 + 4a - 4}}{2} \Leftrightarrow g_a(z) < -1, \ z > 0.$$

Hence we know for any $z_t \in J_2$, we will have

$$|z_{t+1}| = |z_t g_a(z_t)| > |z_t|. \tag{A.11}$$

On the other hand, notice that 0 is in the orbit if and only if $z_0 \notin S_0$, where $S_0$ is defined as

$$S_0 = \bigcup_{n=0}^{\infty} f_a^{(-n)}(0)$$

where $f_a^{-n}(z)$ denotes the set of preimage of $z$ under $f_a^{(n)}$. Note that each preimage is finite and thus $S_0$ is countable. Hence, we know almost surely the orbit will not contain 0, and recall that by Lemma (7) (ii) and $\lim_{t\to\infty} z_t = 0$, we know there are infinitely many $t$ such that $t \in J_2$, and thus (A.11) holds for infinitely many $t$ almost surely. By definition 2, we know $\{|z_t|\}$ has catapults almost surely. ∎

The following theorem indicates that, $f_a$ is chaotic provided that $a > a_*$ where $a_* \in (1, 2)$

**Lemma 9.** *Suppose $1 < a \leq 2$ and $-a \leq z_0 \leq 2$. Then we have*

- *(i) $-a \leq z_t \leq 2$ for any $t$, and $f_a(z)$ has a period-2 point on $[0, 1]$.*

- *(ii) There exists $a_* \in (1, 2)$ such that for any $a \in (a_*, 2)$, $f_a$ is Li-Yorke chaotic, and for any $a \in (1, a_*)$, $f_a$ is not Li-Yorke chaotic.*

- *(iii) If there exists an asymptotically stable orbit and $z_0$ is chosen from $[-a, 2]$ uniformly at random, then the orbit of $z_0$ is asymptotically periodic almost surely.*

*Proof.* The boundedness of $z_t$ is a direct result of Lemma 6 (iii). To prove the rest of (i), we notice that for $a \in (1, 2]$

$$g_a(0)g_a(0g_a(0)) = (1 - 2a)^2 > 1, \ g_a(1)g_a(1g_a(1)) = -a < -1.$$

By continuity of $g_a(zg_a(z))$ we know there exists a point $z_0 \in (0, 1)$ such that $g_a(z_0g_a(z_0)) = 1$. This indicates that $f^{(2)}(z_0) = z_0g_a(z_0g_a(z_0)) = z_0$ but clearly $f_a(z_0) \neq z_0$ since $(0, 1)$ does not contain any fixed point of $f_a$.

To prove (ii), notice that

$$f_1 \left( \frac{1 - 2 \times 1}{3} \right) = \frac{5}{27} < 1 < 2 = f_2 \left( \frac{1 - 2 \times 2}{3} \right).$$

By continuity of $f_a \left( \frac{1-2a}{3} \right)$ (with respect to $a$) there exists $c \in (1, 2)$ such that

$$f_c \left( \frac{1 - 2c}{3} \right) = \frac{(2c - 1)(2c^2 + 7c - 4)}{27} = 1. \tag{A.12}$$

Moreover we have

$$f_c(-c) = -c < \frac{1 - 2c}{3}, \ f_c \left( \frac{1 - 2c}{3} \right) = 1 > \frac{1 - 2c}{3}.$$

Hence by continuity of $f_c(z)$, we can pick $z_0 \in \left( -c, \frac{1-2c}{3} \right)$ such that $f_c(z_0) = \frac{1-2c}{3}$. We have

$$-c < z_0 < \frac{1 - 2c}{3} = f_c(z_0). \tag{A.13}$$

By (A.12), (A.13), and Lemma 6 (i), we have

$$f_c^{(3)}(z_0) = f_c^{(2)} \left( \frac{1 - 2c}{3} \right) = f_c(1) = -c \leq z_0, \tag{A.14}$$

$$f_c(z_0) = \frac{1 - 2c}{3} < 1 = f_c(1) = f_c^{(2)}(z_0). \tag{A.15}$$

Combining (A.13), (A.14), (A.15) we can easily verify that

$$f_c^{(3)}(z_0) \leq z_0 < f_c(z_0) < f_c^{(2)}(z_0).$$

By Theorem B.1 (i.e., Theorem 1 in Li and Yorke (1975)), we know $f_c$ is Li-Yorke chaotic. Moreover, for any $a \in (c, 2]$, we know

$$f_a \left( \frac{1 - 2a}{3} \right) = \frac{(2a - 1)(2a^2 + 7a - 4)}{27} > \frac{(2c - 1)(2c^2 + 7c - 4)}{27} = f_c \left( \frac{1 - 2c}{3} \right) = 1,$$

which together with $f_a(0) = 0 < 1$ implies we can pick $y_0$ such that

$$\frac{1 - 2a}{3} < y_0 < 0, \ f_a(y_0) = 1.$$

Similarly, we have

$$f_a(-a) = -a < \frac{1 - 2a}{3} < y_0, \ f_a \left( \frac{1 - 2a}{3} \right) > 1 > y_0$$

which implies we can pick $x_0$ such that

$$-a < x_0 < \frac{1 - 2a}{3}, \ f_a(x_0) = y_0.$$

Now we know

$$f_a^{(3)}(x_0) < x_0 < f_a(x_0) < f_a^{(2)}(x_0).$$

By Theorem B.1 (i.e., Theorem 1 in Li and Yorke (1975)), we know $f_a$ is Li-Yorke chaotic. Hence, we know $c$ defined in (A.12) satisfies that for any $a \in (c, 2]$, $f_a$ is Li-Yorke chaotic. Hence, we know

$$a_* = \inf_{a \in (1,2)} \{a : f_b \text{ is Li-Yorke chaotic for any } b \in [a, 2].\}$$

where the set is not empty, since we have proven $c$ belongs to the above set. This completes the proof of (ii).

To prove (iii), we notice that if $f_a(z)$ has an asymptotically stable periodic orbit, by Theorem B.2 (i.e., Theorem 2.7 in Singer (1978)) and the fact that $f_a(x)$ has negative Schwarzian derivative at non-critical points (Lemma 4) and we know there exists a critical point $c$ of $f_a(z)$ such that the orbit of $c$ converges to this asymptotically stable orbit. Notice that by Lemma 6 we know $c = 1$ or $\frac{1-2a}{3}$. $c = 1$ can be excluded since $f_a(1) = -a$, and $-a$ is an unstable period-1 point. Hence, we know $c = \frac{1-2a}{3}$ is asymptotically periodic. By Theorems B.3 and B.4 (i.e., Theorem B and Corollary in Nusse (1987)), we know almost surely $z_0$ will not converge to any periodic orbit if $z_0$ is chosen from $[-a, 2]$ uniformly at random. This completes the proof. ∎

**Remarks:**

- See Figure 5(b) for a pair of period-2 points when $a = 1.2$, and Figure 5(c) for a period-3 orbit when $a = 1.6$. The triangle markers denote the periodic points.
- By Theorem B.2 (i.e., Theorem 2.7 in Singer (1978)) and the fact that $-a$ is an unstable period-1 point we know $f_a(z)$ has at most one asymptotically stable periodic orbit.

**Lemma 10.** *Suppose $a > 2$. $z_0$ is chosen from $[-a, 2]$ uniformly at random. Then $\lim_{t\to\infty} |z_t| = +\infty$ almost surely.*

*Proof.* Notice that by Lemma 6 we know

$$f_a\left(\frac{1-2a}{3}\right) = \frac{4a^3 + 12a^2 - 15a + 4}{27} > \frac{(4a^3 + 12a^2 - 15a + 4)|_{a=2}}{27} = 2, \ \forall a > 2,$$

where the inequality holds since $4a^3 + 12a^2 - 15a + 4$ is increasing on $(2, \infty)$. Moreover, we have

$$f_a(z) - z = z(z + a)(z - 2) > 0, \ \forall z \in (2, \infty).$$

Hence we know for the initialization at the critical point $z_0 = \frac{1-2a}{3}$, we have $z_1 > 2$, and the whole sequence is increasing. On the other hand, all fixed points of $f_a(z)$ are no greater than 2, we know $z_t$ will diverge to $+\infty$. For another critical point $z_0 = 1$ we know its orbit converges to the periodic orbit of $z_0 = -a$, which is an unstable period-1 point. Hence, we know from Theorem B.2 (i.e., Theorem 2.7 in Singer (1978)) that there does not exist an asymptotically stable periodic orbit, otherwise the orbit of one critical point must converge to it. Hence, by Theorems B.3 and B.4 (i.e., Theorem B and Corollary in Nusse (1987)) we know $\lim_{t\to\infty} |z_t| = +\infty$ almost surely provided $z_0$ uniformly chosen from $(-a, 2)$, i.e., almost all points in $[-a, 2]$ converge to the absorbing boundary point $+\infty$. ∎

### A.2 Proofs of results in Section 3

*Proof of Theorem 3.1.* Define

$$\alpha^{(t)} := c + \gamma X^\top w^{(t)}, \ \beta := y + \frac{c^2}{2\gamma}, \ \kappa := \eta\gamma \|X\|^2.$$

To prove (i), we observe that

$$\nabla_w g(w; X) = (c + \gamma(X^\top w))X$$

Let weights at time $t$ be $w^{(t)}$. Thus, the gradient descent takes the form

$$w^{(t+1)} = w^{(t)} - \eta(g(w^{(t)}; X) - y)(c + \gamma X^\top w^{(t)})X = w^{(t)} - \eta e^{(t)}\alpha^{(t)}X.$$

Simple calculation gives

$$e^{(t)} = \frac{(\alpha^{(t)})^2}{2\gamma} - \beta \tag{A.16}$$

and

$$\alpha^{(t+1)} = (1 - \eta\gamma \|X\|^2 e^{(t)})\alpha^{(t)} = (1 - \kappa e^{(t)})\alpha^{(t)}.$$

Hence

$$e^{(t+1)} - e^{(t)} = \frac{1}{2\gamma}\left((\alpha^{(t+1)})^2 - (\alpha^{(t)})^2\right) = \left((1 - \kappa e^{(t)})^2 - 1\right)\frac{(\alpha^{(t)})^2}{2\gamma}$$

which together with (A.16) implies

$$\kappa e^{(t+1)} = \kappa e^{(t)}(\kappa e^{(t)} + \beta\kappa)\left(\kappa e^{(t)} - 2\right) + \kappa e^{(t)}.$$

By definition of $a$ and $z_t$ in (3.2) we know $a = \beta\kappa$ and $z_t = \kappa e^{(t)}$. We know (i) holds.

To compute the largest eigenvalue of the Hessain matrix (i.e., the sharpness defined in EoS literature) of the loss in (ii), we notice that the gradient of the loss function takes the form

$$\nabla\ell(w) = (g(w;X) - y)\nabla_w g(w;X).$$

Hence

$$\nabla^2\ell(w) = \nabla_w g(w;X)\nabla_w g(w;X)^\top + (g(w;X) - y)\nabla_w^2 g(w;X) = (\alpha^2 + \gamma e)XX^\top,$$

where we overload the notation and define

$$\alpha = c + \gamma X^\top w, \ e = g(w;X) - y.$$

The sharpness is given by

$$\lambda_{\max}(\nabla^2\ell(w^{(t)})) = ((\alpha^{(t)})^2 + \gamma e^{(t)})\|X\|^2 = (3\gamma e^{(t)} + 2\gamma y + c^2)\|X\|^2 = \frac{3z_t + 2a}{\eta}.$$

∎

*Proof of Theorem 3.2.* The gradient descent takes the form

$$w^{(t+1)} = w^{(t)} - \frac{\eta}{2n}\sum_{i=1}^{n}\nabla\ell_i(w^{(t)}) = w^{(t)} - \frac{\eta}{n}\sum_{i=1}^{n}e^{(t)}(X_i)\alpha^{(t)}(X_i)X_i.$$

Similarly to (A.16), for each error term $e^{(t)}(X_i)$ we have

$$e^{(t)}(X_i) = \frac{(\alpha^{(t)}(X_i))^2}{2\gamma} - \beta(X_i), \tag{A.17}$$

and

$$\alpha^{(t+1)}(X_i) = \gamma X_i^\top w^{(t+1)} + c(X_i)$$

$$= \gamma\left(X_i^\top w^{(t)} - \frac{\eta}{n}\sum_{j=1}^{n}e^{(t)}(X_j)\alpha^{(t)}(X_j)X_i^\top X_j\right) + c(X_i)$$

$$= \alpha^{(t)}(X_i) - \frac{\gamma\eta}{n}\sum_{j=1}^{n}e^{(t)}(X_j)\alpha^{(t)}(X_j)X_i^\top X_j$$

$$= \alpha^{(t)}(X_i) - \frac{\gamma\eta}{n}\sum_{j=1}^{n}\left(\frac{\alpha^{(t)}(X_j)^3}{2\gamma} - \beta(X_j)\alpha^{(t)}(X_j)\right)X_i^\top X_j$$

We overload the notation and set

$$\mathbf{X} = (X_1, ..., X_n)^\top, \ \#(\mathbf{X}) = (\#(X_1), ..., \#(X_n))^\top, \ \forall\# \in \{\alpha^{(t)}, e^{(t)}, a, \beta\}.$$

We can obtain

$$\alpha^{(t+1)}(\mathbf{X}) = \alpha^{(t)}(\mathbf{X}) - \frac{\eta}{n}\mathbf{X}\mathbf{X}^\top\left(\frac{\alpha^{(t)}(X)^3}{2} - \gamma\beta(X)\odot\alpha^{(t)}(X)\right), \tag{A.18}$$

where $\odot$ denotes the Hadamard product.

As $\mathbf{X}\mathbf{X}^\top = \mathrm{diag}(\|X_1\|^2, ..., \|X_n\|^2)$, we can rewrite (A.18) as the following non-interacting version for each data point:

$$
\begin{aligned}
\alpha^{(t+1)}(X_i) &= \alpha^{(t)}(X_i) - \frac{\eta\|X_i\|^2}{2n}\left(\alpha^{(t)}(X_i)^3 - 2\gamma\beta(X_i)\alpha^{(t)}(X_i)\right) \\
&= \left(1 - \frac{\gamma\eta\|X_i\|^2}{n}e^{(t)}(X_i)\right)\alpha^{(t)}(X_i).
\end{aligned}
$$

This together with (A.17) implies

$$
\begin{aligned}
e^{(t+1)}(X_i) - e^{(t)}(X_i) &= \frac{1}{2\gamma}\left((\alpha^{(t+1)}(X_i))^2 - (\alpha^{(t)}(X_i))^2\right) \\
&= \left(-\frac{2\gamma\eta\|X_i\|^2}{n}e^{(t)}(X_i) + \frac{\gamma^2\eta^2\|X_i\|^4}{n^2}(e^{(t)}(X_i))^2\right)\left(e^{(t)}(X_i) + \beta(X_i)\right) \\
&= \kappa_n(X_i)e^{(t)}(X_i)\left(\kappa_n(X_i)e^{(t)}(X_i) - 2\right)\left(e^{(t)}(X_i) + \beta(X_i)\right)
\end{aligned}
$$

By definition of $z_i^{(t)}$ and $a_i$ we know

$$
z_i^{(t+1)} = z_i^{(t)}(z_i^{(t)} + a_i)(z_i^{(t)} - 2) + z_i^{(t)} = f_{a_i}(z_i^{(t)}).
$$

The sharpness is given by

$$
\begin{aligned}
\nabla^2\ell(w^{(t)}) &= \frac{1}{n}\sum_{i=1}^n\left(\nabla_w g(w^{(t)}; X_i)\nabla_w g(w^{(t)}; X_i)^\top + (g(w^{(t)}; X_i) - y_i)\nabla_w^2 g(w^{(t)}; X_i)\right) \\
&= \frac{1}{n}\sum_{i=1}^n\left((\alpha^{(t)}(X_i))^2 + \gamma e^{(t)}(X_i)\right)X_i X_i^\top \\
&= \frac{1}{n}\sum_{i=1}^n(3\gamma e^{(t)}(X_i) + 2\gamma y_i + c^2(X_i))X_i X_i^\top.
\end{aligned}
$$

Therefore we know

$$
\nabla^2\ell(w^{(t)})X_i = \frac{1}{n}(3\gamma e^{(t)}(X_i) + 2\gamma y_i + c^2(X_i))\|X_i\|^2 X_i = \frac{3z_i^{(t)} + 2a_i}{\eta}X_i, \quad \text{for all } 1 \le i \le n.
$$

which means we find $n$ eigenvalues and eigenvectors pairs $\left(\frac{3z_1^{(t)}+2a_1}{\eta}, X_1\right), ..., \left(\frac{3z_n^{(t)}+2a_n}{\eta}, X_n\right)$.
Note that $\nabla^2\ell(w^{(t)})$ is a sum of $n$ rank-1 matrices, and we have found $n$ orthogonal eigenvalues.
Hence we know $\lambda_{\max}(\nabla^2\ell(w^{(t)})) = \max_{1\le i\le n}\frac{3z_i^{(t)}+2a_i}{\eta}$. This completes the proof. ∎

*Proof of Theorem 3.3.* Define

$$
\mathbf{A}^{(t)} = \frac{2\eta}{\sqrt{m}dn}\sum_{j=1}^n e_j^{(t)}X_j X_j^\top.
$$

Note that we have

$$
\nabla\ell_j^{(t)}(\mathbf{U}^{(t)}) = \left(\frac{1}{\sqrt{m}d}\sum_{i=1}^m(X_j^\top u_i^{(t)})^2 - y_j\right)\left(\frac{2}{\sqrt{m}d}X_j X_j^\top\mathbf{U}^{(t)}\right) = \frac{2}{\sqrt{m}d}e_j^{(t)}X_j X_j^\top\mathbf{U}^{(t)}.
$$

This implies that the gradient descent update takes the form

$$
\mathbf{U}^{(t+1)} = \mathbf{U}^{(t)} - \frac{\eta}{n}\sum_{j=1}^n\nabla\ell_j^{(t)}(\mathbf{U}^{(t)}) = \mathbf{U}^{(t)} - \frac{2\eta}{\sqrt{m}dn}\sum_{j=1}^n e_j^{(t)}X_j X_j^\top\mathbf{U}^{(t)} = \left(I - \mathbf{A}^{(t)}\right)\mathbf{U}^{(t)}.
$$

Also we have

$$e_j^{(t+1)} - e_j^{(t)} = \frac{1}{\sqrt{md}} \sum_{i=1}^{m} \left( (X_j^\top u_i^{(t+1)})^2 - (X_j^\top u_i^{(t)})^2 \right)$$

$$= \frac{1}{\sqrt{md}} X_j^\top \left( \mathbf{U}^{(t+1)}(\mathbf{U}^{(t+1)})^\top - \mathbf{U}^{(t)}(\mathbf{U}^{(t)})^\top \right) X_j$$

and

$$\mathbf{U}^{(t+1)}(\mathbf{U}^{(t+1)})^\top = \left( I - \frac{2\eta}{\sqrt{md}dn} \sum_{j=1}^{n} e_j^{(t)} X_j X_j^\top \right) \mathbf{U}^{(t)}(\mathbf{U}^{(t)})^\top \left( I - \frac{2\eta}{\sqrt{md}dn} \sum_{j=1}^{n} e_j^{(t)} X_j X_j^\top \right).$$

Hence we know

$$e_j^{(t+1)} - e_j^{(t)}$$
$$= \frac{1}{\sqrt{md}} \left( X_j^\top \mathbf{A}^{(t)} \mathbf{A}^{(t)} (\mathbf{A}^{(t)})^\top \mathbf{A}^{(t)} X_j - 2 X_j^\top \mathbf{A}^{(t)} \mathbf{U}^{(t)} (\mathbf{U}^{(t)})^\top X_j \right)$$
$$= \frac{1}{\sqrt{md}} \left( \frac{4\eta^2}{md^2n^2} (e_j^{(t)})^2 \|X_j\|^4 X_j^\top \mathbf{U}^{(t)}(\mathbf{U}^{(t)})^\top X_j - \frac{4\eta}{\sqrt{md}dn} e_j^{(t)} \|X_j\|^2 X_j^\top \mathbf{U}^{(t)}(\mathbf{U}^{(t)})^\top X_j \right)$$
$$= \left( \frac{4\eta^2 \|X_j\|^4}{md^2n^2} (e_j^{(t)})^2 - \frac{4\eta \|X_j\|^2}{\sqrt{md}dn} e_j^{(t)} \right) \left( e_j^{(t)} + y_j \right),$$

where the second equality uses $\mathbf{X}\mathbf{X}^\top = \text{diag}(\|X_1\|^2, ..., \|X_n\|^2)$. By definition of $z_i^{(t)}$ and $a_i$ we know

$$z_i^{(t+1)} = f_{a_i}(z_i^{(t)}).$$

Hence we know the training dynamics of this model can be captured by the cubic map as well. ∎

## B  AUXILIARY RESULTS

**Theorem B.1** (Theorem 1 in Li and Yorke (1975)). *Let $I$ be a compact interval and let $f : I \to I$ be continuous. Assume there is a point $a \in I$ for which the points $b = f(a)$, $c = f^{(2)}(a)$ and $d = f^{(3)}(a)$ satisfy*

$$d \leq a < b < c \; (or \; d \geq a > b > c).$$

*Then $f$ is Li-Yorke chaotic.*

**Theorem B.2** (Theorem 2.7 in Singer (1978)). *Let $I$ be a compact interval and let $f : I \to I$ be a three times continuously differentiable function. If the Schwarzian derivative of $f$ satisfies*

$$\mathsf{S}f(x) = \frac{f'''(x)}{f'(x)} - \frac{3}{2} \left( \frac{f''(x)}{f'(x)} \right)^2 < 0 \; for \; all \; x \in I \; with \; f'(x) \neq 0.$$

*Then the stable set of every asymptotically stable orbit of $f$ contains a critical point of $f$.*

**Theorem B.3** (Theorem B in Nusse (1987)). *Let $I$ be an interval and let $f : I \to I$ be a three times continuously differentiable function having at least one aperiodic point on $I$ and satisfying:*

- *(i) $f$ has a nonpositive Schwarzian derivative, i.e.,*

$$\mathsf{S}f(x) = \frac{f'''(x)}{f'(x)} - \frac{3}{2} \left( \frac{f''(x)}{f'(x)} \right)^2 \leq 0 \; for \; all \; x \in I \; with \; f'(x) \neq 0.$$

- *(ii) The set of points, whose orbits do not converge to an (or the) absorbing boundary point(s) of $I$ for $f$ is a nonempty compact set.*
- *(iii) The orbit of each critical point for $f$ converges to an asymptotically stable periodic orbit of $f$ or to an (or the) absorbing boundary point(s) of $I$ for $f$.*

- *(iv) The fixed points of $f^{(2)}$ are isolated.*

*Then we have*

- *(1) The set of points whose orbits do not converge to an asymptotically stable periodic orbit of $f$ or to an (or the) absorbing boundary point(s) of $I$ for $f$ has Lebesgue measure $0$;*
- *(2) There exists a positive integer $p$ such that almost every point $x$ in $I$ is asymptotically periodic with $f^{(p)}(x) = x$, provided that $f(I)$ is bounded.*

**Theorem B.4** (Corollary in Nusse (1987)). *Assume that $f : \mathbb{R} \to \mathbb{R}$ is a polynomial function having at least one aperiodic point and satisfying the following conditions:*

- *(i) The orbit of each critical point of $f$ converges to an asymptotically stable periodic orbit of $f$ or to an (or the) absorbing boundary point(s) for $f$;*
- *(ii) Each critical point of $f$ is real.*

*Then $f$ satisfies the assumptions (i)-(iv) of Theorem B.3.*

## C  EXPERIMENTAL INVESTIGATIONS (CONTINUED FROM SECTION 4)

Due to space limitation, we provide additional experimental results in this section.

### C.1  ADDITIONAL EXPERIMENTS FOR THE ORTHOGONAL CASE

For this section, we follow the same experimental setup as described in Section 4.1. Only the hidden-layer width is changed. Specifically, in Figures 6 and 7 we plot the training loss, sharpness of training loss and the trajectory-averaging training in various phases.

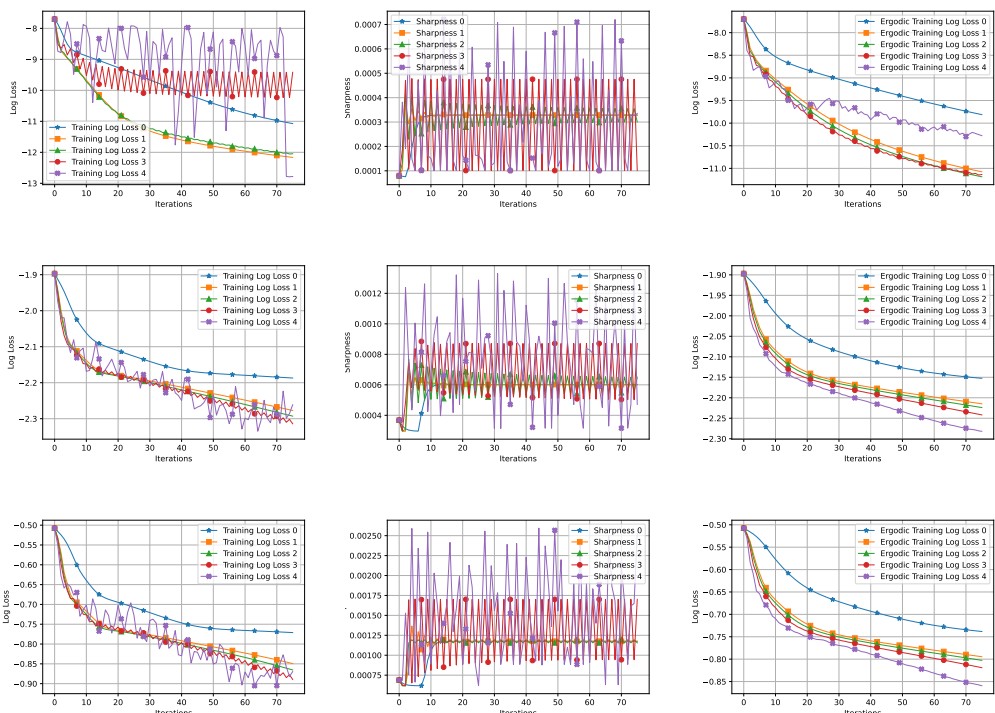

Figure 6: Hidden-layer width =5, with orthogonal data points. Rows from top to bottom represent different levels of noise – mean-zero normal distribution with variance $0, 0.25, 1$. The vertical axes are in log scale for the training loss curves. The second column is about the sharpness of the training loss functions.

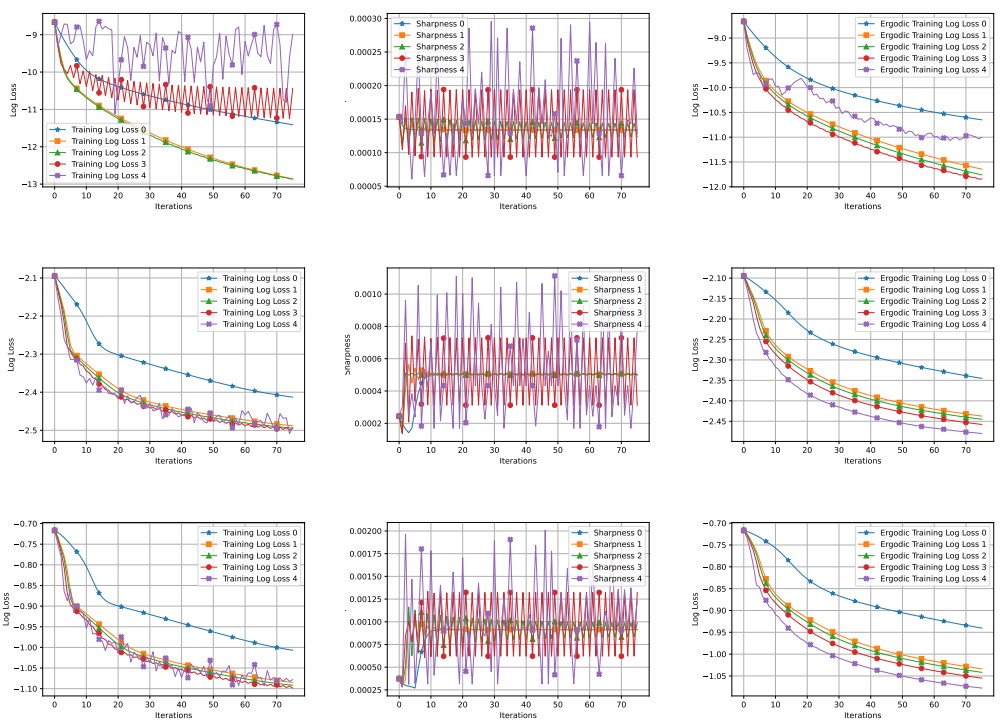

Figure 7: Hidden-layer width =10, with orthogonal data points. Rows from top to bottom represent different levels of noise – mean-zero normal distribution with variance $0, 0.25, 1$. The vertical axes are in log scale for the training loss curves. The second column is about the sharpness of the training loss functions.

## C.2 NON-ORTHOGONAL DATA

We next investigate the case when orthogonality condition does not hold. The setup is the same as described in Section 4.1 except that $n = 5000$ and each entry of the data matrix $X \in \mathbb{R}^{n \times d}$ is now sampled from a standard normal distribution. We also generate 500 data points from the same distribution for testing. Note that our theory in this work is only applicable for orthogonal data. hence, for these experiments with non-orthogonal data, we first tune the step-size to be as large as possible, say $\eta_{\max}$, so that the training does not diverge and then run the experiments for $\frac{i+1}{5}\eta_{\max}$ with $i = 0, ..., 4$. Hence, the step-sizes for loss and sharpness curves $0, 1, 2, 3, 4$ are chosen to be $10, 20, 30, 40, 50$ for $m = 5, 10$ and $12, 24, 36, 48, 60$ for $m = 25$.

In Figures 8, 9 and 10 we plot the training loss and the testing loss (with and without ergodic trajectory averaging) in log scale. Notably different phases (including the periodic and catapult phases) characterized theoretically for the case of orthogonal data, also appear to be present for the non-orthogonal case. We also make the following intriguing conclusions:

- As a general trend, training roughly in the catapult phase and predicting without doing the ergodic trajectory averaging appears to have the best test error performance.
- In some cases (especially the one with high noise variance), when testing after training in the periodic phase, the test error goes down rapidly in the initial few iterations. Correspondingly, ergodic trajectory averaging after training in the periodic phase, helps to obtain better test error decay compared to ergodic trajectory averaging after training in the catapult phase. However, as mentioned in the previous point, training roughly in the catapult phase and predicting without doing the ergodic trajectory averaging performs the best.
- As discussed in Lim et al. (2022), in various cases, artificially infusing control chaos help to obtain better test accuracy. Given our empirical observations and the results in Lim et al. (2022), it is interesting to design controlled chaos infusion in gradient descent and perform ergodic training averaging to obtain stable and improved test error performance.

Obtaining theoretical results corroborating the above-mentioned observations is challenging future work.

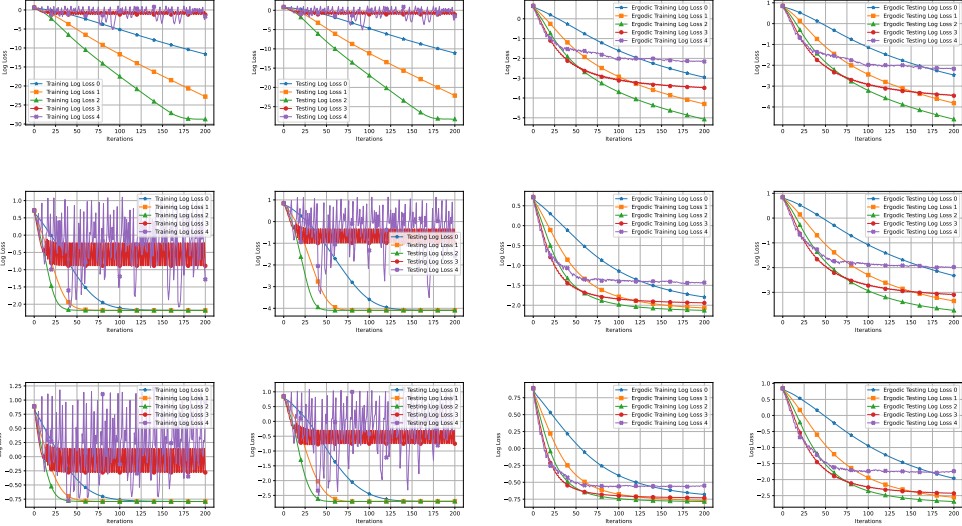

Figure 8: Hidden-layer width=5, with non-orthogonal data points. Rows from top to bottom represent different levels of noise – mean-zero normal distribution with variance $0, 0.25, 1$. The vertical axes are in log scale for loss curves.

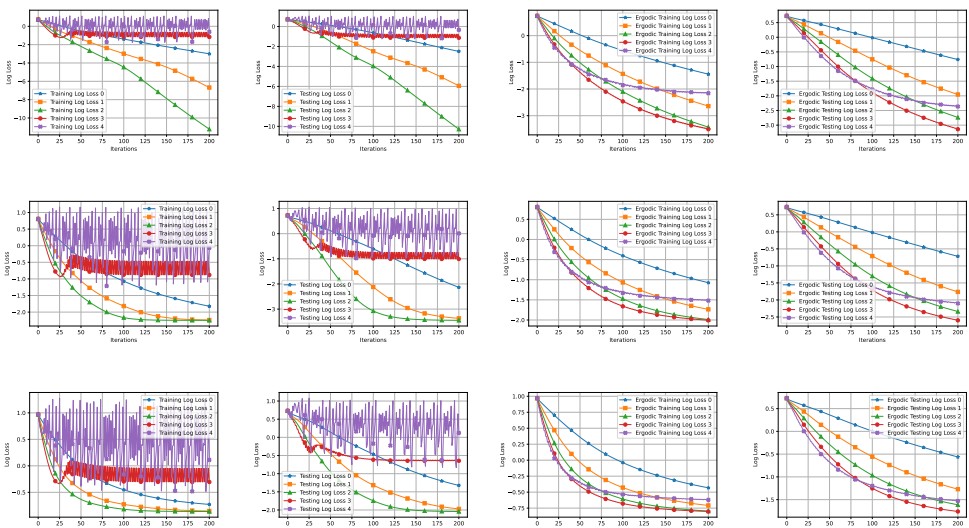

Figure 9: Hidden-layer width=10, with non-orthogonal data points. Rows from top to bottom represent different levels of noise – mean-zero normal distribution with variance $0, 0.25, 1$. The vertical axes are in log scale for loss curves.

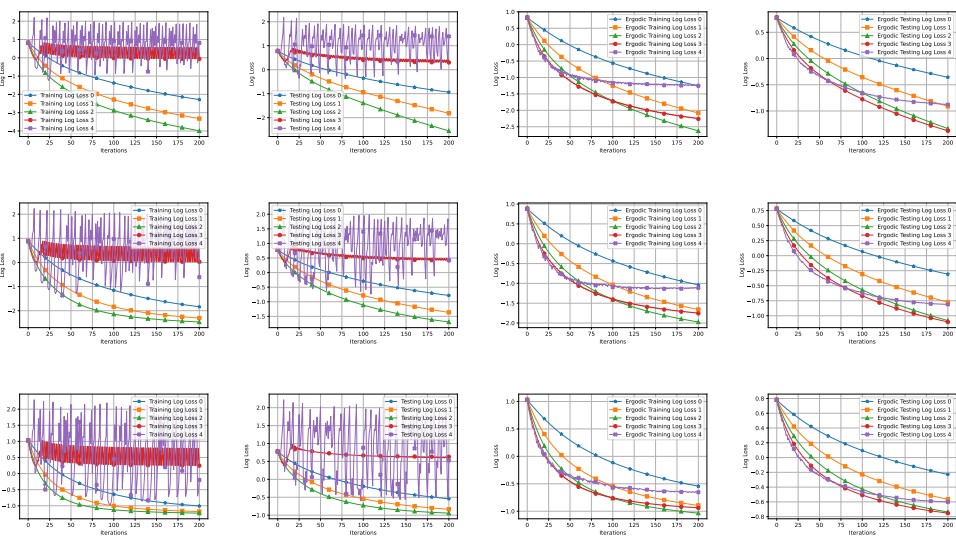

Figure 10: Hidden-layer width=25, with non-orthogonal data points. Rows from top to bottom represent different levels of noise – mean-zero normal distribution with variance $0, 0.25, 1$. The vertical axes are in log scale for loss curves.

### C.3 TWO-LAYER NEURAL NETWORK WITH RELU

While our main focus in this work is for quadratic activation functions, it is also instructive to examine the dynamics with other activation function, in particular the ReLU activation. Hence, we follow the experimental setup from Section C.2, except that the activation function is now ReLU and repeat our experiments. For this case, the step-sizes manually chosen to be $60, 120, 180, 240, 300$ for loss/sharpness curves $0, 1, 2, 3, 4$, respectively.

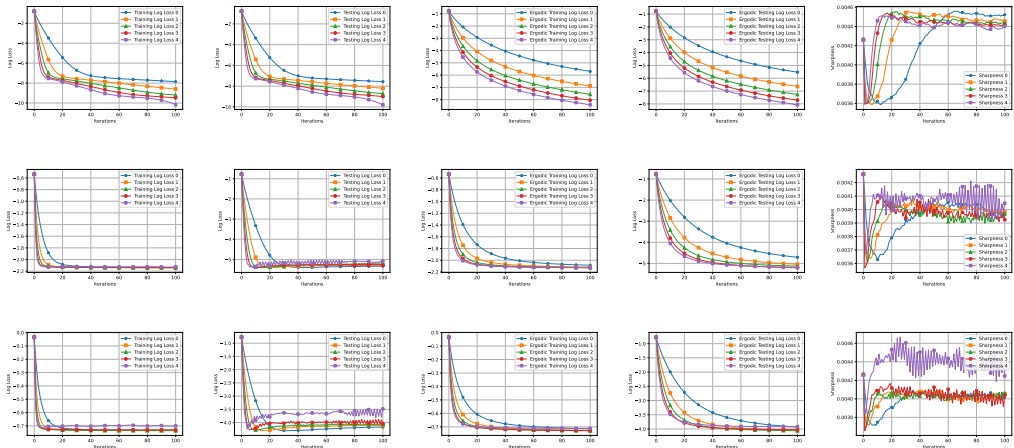

Figure 11: Hidden-layer width=5 with ReLU activation. Rows from top to bottom represent different levels of noise – mean-zero normal distribution with variance 0, 0.25, 1. The vertical axes are in log scale for loss curves. The last column is about the sharpness of the training loss functions.

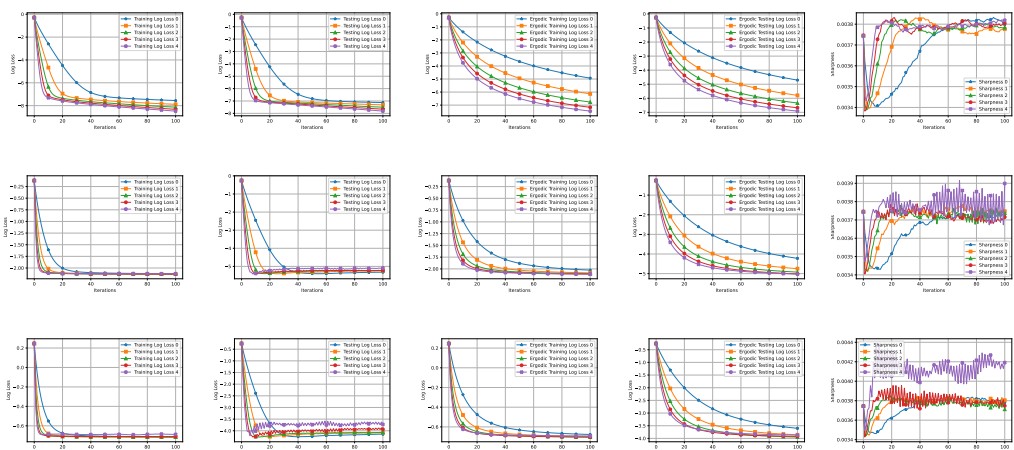

Figure 12: Hidden-layer width=10 with ReLU activation. Rows from top to bottom represent different levels of noise – mean-zero normal distribution with variance 0, 0.25, 1. The vertical axes are in log scale for loss curves. The last column is about the sharpness of the training loss functions.

From Figures 11 and 12, (in particular from the sharpness plots), we observe various non-monotonic patterns, roughly including periodic and chaotic patterns. Obtaining a precise theoretical characterization of the training dynamics for this setting is extremely interesting as future work.

