# OpenReview forum: "From Stability to Chaos: Analyzing Gradient Descent Dynamics in Quadratic Regression"
_ICLR.cc/2024/Conference — ICLR 2024 Conference Withdrawn Submission_

### Official Review · Reviewer_BLuU · 2023-10-29

**Soundness:** 3 good
**Presentation:** 2 fair
**Contribution:** 1 poor
**Rating:** 3
**Confidence:** 5

**Summary:**

In this work the authors analyze a cubic dynamical system and show its distinct phases, from convergent to oscillatory to chaotic. They then show that this dynamical system captures the behavior of one-hidden layer neural networks with quadratic activation and fixed readout layer of all 1s. They provide numerical evidence for all the phases, and also provide evidence that in the chaotic phase, averaging over the trajectories gives better predictions.

**Strengths:**

The identification and analysis of the simplified dynamical system is very sound. This paper does a good job of bringing in appropriate ideas from dynamical systems analysis into analysis of machine learning algorithms. The different regions are clearly defined, and defined clearly. The analysis is successfully extended to a simple neural network setup with multiple datapoints. The work does a good job positioning itself properly in the literature.

**Weaknesses:**

Overall, I have concerns about the applicability of the analysis to more general ML models, even those studied theoretically. It seems that the analysis relies heavily on both the factorizability of the specific models, as well as the quadratic activation function.

In addition, it's not clear to me which of the phases are relevant even for the training of the models discussed. Additional thoughts on this point are in the "Questions" section.

Most of the figures are hard to parse; more specifics can be found in the "Questions" section.

The discussion in Section 4.2 relating the dynamics to SGD are simply incorrect. See for example [1] for ways in which SGD in even a simple loss behaves quite differently from full batch GD.



[1] https://proceedings.neurips.cc/paper_files/paper/2022/hash/efcb76ac1df9231a24893a957fcb9001-Abstract-Conference.html

**Questions:**

Figure 1 is very small and hard to read. All the figure axis labels and ticks are too small to parse. Additionally for Figure 2, the concept of the diagram should be explained. For figure 4, it would be useful to indicate the EOS on the sharpness plots.

One suggestion for Section 2: there are a lot of definitions, which may or may not be familiar to readers interested in the paper. It might be helpful if the authors could write a small appendix with simple examples of dynamical systems/fixed points that meet the various definitions/rely on the various definitions. This could help a reader unfamiliar with dynamical systems analysis gain intuition before trying to dive into the main results.

For the different phases: how are the sharpness and loss dynamics related to the phases and known phenomenology like Edge of Stability? Ostensibly this information is available in Figure 4, but I find it very hard to parse in its current form. More explicit discussion of these points, and either improved points (or maybe some sort of table of converged values?) would be valuable. It is hard to understand the limiting behaviors of the exact forms of the final sharpness/loss in the current iteration.

---

### Official Review · Reviewer_fNyy · 2023-10-31

**Soundness:** 3 good
**Presentation:** 2 fair
**Contribution:** 1 poor
**Rating:** 3
**Confidence:** 3

**Summary:**

This paper studies the training trajectory of GD for quadratic regression in the framework of dynamical system. A model describing the dynamics of training is studied, and five phases (monotonic, catapult, periodic, chaotic, divergent) are identified. Most of the phase-transition points are located quantitatively. It is shown that this model can describe a generalized phase retrieval problem and training a restricted two-layer network with quadratic activation function. Numerical results confirm the existence of the phases as well as highlighting the necessity of ergodic trajectory averaging, a technique proposed in this paper to stabilize the performance of the trained model.

**Strengths:**

This paper studies the training dynamics in the framework of dynamical system and carries out a thorough investigation of the dynamical system at an unprecedented level. The mathematical results are presented clearly.

**Weaknesses:**

Although this paper could be interesting for specialist in dynamical system, its relevance in the context of machine learning could be limited:
* The dynamical system analyzed in the paper could be reduced from two problems, both with strong constraint (orthogonal data or fixed weights in the neural network). It appears to me that this is not the most representative problem in machine learning in general.
* It appears to me that the phases identified in this paper are related to the convergence of gradient descent. However, there is no comparison of the quantitative result in this paper to the ones in other papers studying the same problem using different methods (for example, papers surveyed in section 1.1, General results.) This makes the analysis in this paper isolated.
* The numerical experiment is performed using synthesized data instead of real ones.

Also, although the mathematical results in the main texts are clear, I find the experimental results hard to interpret. Notably:
* The performance of ergodic trajectory averaging is evaluated in synthesized data. It appears to me that this technique could be beneficial in realistic machine learning tasks. However, no such experiment is performed.
* I have problems interpreting Figure 4. I am not sure how I should interpret the legend in the figures, and I am not sure what message I should get from this figure.

**Questions:**

* The mathematical result in the paper indicates that the trajectory of gradient descent depends heavily on the interplay between the learning rate and the initialization. However, in practice, it appears to me that initialization and learning rate scheme are hardly considered together. I wonder if this paper provides any insight into the relation between the two in practice.
* I would like to confirm with the authors that the word ‘step-size’ in the paper indeed means ‘learning rate’. If yes, it appears to me that the learning rate in the experiments is really large in the experiment presented by Figure 3 (36, 48, 60).
* I wonder if the authors have any comments or objections regarding the weaknesses of the paper in this review.

---

### Official Review · Reviewer_MkpK · 2023-10-31

**Soundness:** 3 good
**Presentation:** 3 good
**Contribution:** 2 fair
**Rating:** 5
**Confidence:** 3

**Summary:**

This paper investigates the dynamics of gradient descent with large constant step sizes in two models: generalized phase retrieval and neural network with quadratic activation. Specifically, assuming orthogonal data, the dynamics of gradient descent are modeled as iterations of a cubic polynomial parameterized by the step size. A detailed bifurcation analysis is provided to determine the different training phases of gradient descent. Additionally, experiments without assuming orthogonal data are conducted, and the experimental results demonstrate that the theoretical findings under orthogonal data assumption hold generally. The generalization performance is also considered empirically, and experimental results show that an ergodic trajectory averaging stabilizes the test error. The techniques and results of this paper are interesting and have left many directions for future study.

**Strengths:**

1.The dynamics of gradient descent with large step sizes is an important topic, and this paper provides new investigations on this problem by offering a detailed analysis of several simple models.

 2.The proof idea of theoretical results by modeling the dynamics as iteration of cubic polynomial is clear and interesting.

  3.Additional experimental results demonstrate that the findings can also be applied in a more general context.

**Weaknesses:**

The assumption of orthogonal data is an obvious weakness that weakens the theoretical contribution of the current paper. Even I believe it is reasonable to leave this as a future problem at the current stage, it is not clear that the technical barrier will not affect the results of dynamical systems in Theorem 2.1. Let me be specific.

The orthogonality assumption can be translated into following statement: let $F(\textbf{X})$ be a mapping from $\mathbb{R}^{dn}$ to $\mathbb{R}^{\frac{n^2-n}{2}}$ defined by
$$F_1(\textbf{X})=X_1^{\top}X_2$$...$$F_{\frac{n^2-n}{2}}(\textbf{X})=X_{n-1}^{\top}X_n$$
i.e., the upper triangle of $\textbf{X}\textbf{X}^{\top}$ without diagonals. The orthogonality is equivalent to $\textbf{X}\in F^{-1}(0)$ where $0\in\mathbb{R}^{\frac{n^2-n}{2}}$. I have not found a detailed discussion on the relation between $d$ and $n$ (or I have missed it). Intuitively it is a crucial aspect to the results of this paper. Roughly speaking, let's say $dn>\frac{n^2-n}{2}$ (i.e. $d>\frac{n-1}{2}$), then what has been proved in Theorem 2.1 holds only if the data set is from a measure zero set ($F^{-1}(0)$ is a lower dimensional manifold, something like that, I did not check the regularity at $0$). I am not sure if we can perturb the current argument from a measure zero set, since analyzing interacting dynamics is a challenging and unexplored direction (Xu et al, 2022).

Therefore, from this perspective, we can only rigorously state: The main results holds if $d>\frac{n-1}{2}$ and the data matrix lies on a measure set, which is generically impossible.

More importantly, even we believe what is indicated in this paper, the requirement of $d>\frac{n-1}{2}$ is unnatural. The size of training set can be arbitrarily large and the dimension of data point is supposed to be fixed as long as the training task is determined. Thus the assumption of orthogonality is expected to capture the case of $n>>d$, but then there seems to be much more work ahead since the equations of defining $\textbf{X}\in F^{-1}(0)$ is over-determined and a careful analysis is needed. Unfortunately, this is completely ignored in this paper and the experiment does not capture this case since $d=100$ and $n=80$, or in the appendix, the non-orthogonal data set has not provided the choice of $d$.

Above observation is crucial in evaluating the importance of the results in this paper, from a machine learning perspective. Note that these concerns might be resolved or argued to be trivial, that will be great if true. The techniques used in this work is interesting and the paper has provided a profound connection between dynamical system and training process.

**Questions:**

1. Corollary 1 doesn't distinguish catapult, periodic, and chaos behaviors. However, from the experiments it seems both these three patterns can appear. How can we determine the range of step sizes that can differentiate these three behaviors ?

     2. In Theorem 3.3, it appears that the non-decreasing behavior of gradient descent can only occur in this model when the step size is on the order of $O(\sqrt{m}dn)$, which seems to be a very large value. What are the step sizes used in related research? Can the results in the current paper be compared to them? Additionally, as shown in C.1, several different step sizes are chosen to demonstrate various dynamics behaviors. Do these step sizes conform to the theoretical results?

     3. It is claimed that training in the phases of periodicity and chaos provides the highest test accuracy in the conclusion part. However, in figure 8 it is shown that the testing losses of periodicity and chaos are larger than the decreasing phase. Am I missing something here ?

4. In C.2 non-orthogonal data, what is $d$ in this experiment?

---

### Official Review · Reviewer_Ebrh · 2023-11-01

**Soundness:** 3 good
**Presentation:** 3 good
**Contribution:** 2 fair
**Rating:** 6
**Confidence:** 3

**Summary:**

In this work, the authors are primarily interested in investigating the dynamics of gradient descent in quadratic or second-order regression models. It is worth noting that the analysis has been carried out for any fixed step size since their main goal is to describe how the dynamics differ across various step-size. To achieve this, they analyze a specific cubic map and delineate its five distinct phases. In the second part of their work, they integrate this analysis into three regression models.

**Strengths:**

The paper is generally well-written; however, there are sporadic grammatical mistakes that hinder the ease of reading. Undeniably, the main contribution of this work lies in the analysis of the cubic map presented in Section 2. The analysis of their cubic map is clearly written and presented.

The paper also features a substantial number of references, which helps the reader to understand the motivation and the related work, and also shows the authors' familiarity with the literature.

**Weaknesses:**

In Section 3, the models they consider may be somewhat simplistic, and theirs further simplifying assumptions (orthogonality and quadratic activation functions) limit the introduction of complexity. For example, it would have been better if they have considered more “realistic” activation functions. It gives the impression that these examples are carefully taken to fit the model (cubic map).

I believe that this work has some merit, but I am unsure whether it exceeds the threshold for acceptance at ICLR.

**Questions:**

Regarding the figures, I am not entirely sure of the key takeaway points. Can you draw any conclusions about generalization (as mentioned in the introduction as the main motivation) or faster convergence rates? It appears that an exposition of the experimental results is missing.

---

### Official Review · Reviewer_qRAi · 2023-11-01

**Soundness:** 3 good
**Presentation:** 2 fair
**Contribution:** 2 fair
**Rating:** 5
**Confidence:** 3

**Summary:**

The submission presents an analysis of the dynamics of gradient descent on non-linear problems through a simplified 1-d model. The main results show that, depending on the step-size, the loss can converge with monotonic decrease, non-monotonic decrease, oscillate periodically, exhibit chaotic behavior, or diverge. The results are illustrated through examples on quadratic regression and fitting the first layer of a 2-layer neural network with quadratic activations under the simplifying assumption that the features are orthogonal. The paper presents experimental evidence that similar dynamics appear without this simplifying assumption.

**Strengths:**

From my reading, the main contribution of the paper is to show that the dynamics of training can be complicated even on a simple model. Specifically, that the full range of behavior observed in practice (phases 1-5) are reproducible on this toy example, not only in practice but provably so. I believe this contribution is novel and would be of interest to the part of the community interested in training dynamics and dynamical systems.

**Weaknesses:**

I think this could be a strong submission, but its current message is somewhat unclear, specifically in its relation to other empirical work. My main concerns are about
- Its main message; the motivation for and the significance of the contribution for a general ML audience and empirical researchers
- And possibly related, the relationship between the submission and concepts originating from more empirical works, such as the edge of stability and catapults.

The paragraphs below provide specific examples of where I think the submission lacks in clarity, in the hope this feedback can help a revision to better convey the intended message of the submmission. _I do not expect a detailled response to the points below. I would rather the discussion period focus on clarifying the motivation and significance aspects raised in the questions._
If a revision adresses those issues, I will raise my score.

---

**Motivation and significance for a more general audience**

The introduction conveys the message that the dynamics of gradient descent on non-linear models are poorly understood, and that the quadratic model has been used in prior work. The related work seems sufficiently comprehensive. But I find it difficult to articulate what broader problem the submission is making progress towards.

The related work section focuses on highlighting the difference between existing work and this submission ("The result are not applicable to quadratic regression", do not "formally prove the existence of chaos", do not "characterize the 5 phases", "not applicable to the cubic map"). This does not show that the submission is "solving a problem", but rather presents a different analysis on a different settings. It might be an interesting mathematical result, but it is not clear what impact it would have on the broader ML community.

For example, from an optimization perspective, it is not clear that the later settings (periodic, chaotic and divergent) are relevant, as it's not converging regardless. Part of the broader goal might be the connections with unstable convergence, edge of stability and catapults, but if so those relationships are not clear after my reading.


**Defining EOS and Catapults, specifying what is meant by similarity**
Places where the text states a similarity or relations between concepts, for example "Various patterns like catapult (also related to edge of stability)" (§1.1, pararagraph 2), would be improved by explicitly stating what the similarity/relation is. Is the wording referring to a specific relationship, or using it to convey the message that both exhibit non-monotonic behavior? It is not clear to me what the relation is, especially as those concepts come from empirical observations rather than formal definitions.

The working definition of catapult used by Lewkowycz et al. includes the idea that the sharpness decreases after a catapult. The definition of Edge of stability includes the idea that the sharpness rises to 2/step-size. To better place the submission in context and make specific claims, the submission should state such working definitions, as not every reader might hold the same understanding.

Such a working definition is somewhat implied by the use of the wording "catapult phase" for the convergent but non-monotonic phase, but I would encourage the more neutral `non-monotonic phase`. Although this phrasing mirrors the one used by Lewkowycz et al., it is implying that the defining behavior of this phase is the presence of catapults, which is not established. Calling it `non-monotonic` and stating that it this is the phase we can expect catapults to exist would be more accurate. Unless, of course, the point that the authors want to make is that this is the _definition_ of catapults, but this does not appear to be the case from my reading.


Another point where what is meant by "relation" should be clarified is in"the chaotic dynamics (and related stochasticity)" (§1.1, par. General results) and "A main take-away from our analysis and experiments so far is that [GD] behaves like [SGD], except that the randomness here is with respect to the orbit it converges to" (§4.2) use "stochastic" and "random" in a way I do not recognise, as the mechanisms studied in the submission are deterministic.


**Terminology**
The submission would benefit from being more careful in its choice of terminology. The two more noticeable examples were
- "large-order step-size", which is used in multiple places. The specific phrasing makes me think there might be a special definition of what "large-order" means, as opposed to the simpler "large", but no definition is provided. If there is no special meaning, I would suggest using "large".
- "Phase", which seems to refer to the behavior of the algorithm with different step-sizes. This might make sense to an audience coming from dynamical system or who have read the work of Lewkowycz et al. recently, but this term is overloaded. It is fine when used as in "phase transition", but in "training phase" it could be understood as referring to different times of one training procedure. I would suggest not using "training phase" and making this usage clear on first use for a more general ML audience.


**Misc.**
- The second and third paragraphs conflate two settings. The statements  "Very much violating the classical case" (§1, paragraph 3) and "In fact, the necessity for larger step-sizes to expedite convergence and the ensuing chaotic behavior has also been observed empirically ... by Van Den Doel and Ascher (2012)" (§1.1, paragraph 2) appear misleading in the context of the submission, or should at least be modified to specify "the necessity _for step-size schedules to include large step-sizes_". As it stands, the statement applies to GD generally, including the setting studied in the paper, and it is not clear that observations in one setting transfer to the other(the two settings being [step-size _schedules_ on (classes of functions such as smooth/strongly convex functions that include) quadratics leading to (almost) linear map updates] vs [non-linear maps with constant step-size]).
- The illustrations need improvements. The font size is too small to be readable. The figures with multiple lines (Fig. 3, 4) would benefit from having different line styles to be readable when printed.

**Questions:**

The point I would appreciate a clarification from the authors is on the motivation and significance for a more general audience, and the relationship between the submission and the empirical work on EOS/catapults.

This might be my bias coming from an optimization background, but it is not clear to me what a member of the general ML audience, as opposed to a specialist in chaotic dynamical systems, should take away from the main findings of the paper.